

# A Frontal Ablation Dataset for 49 Tidewater Glaciers in Greenland

Dominik Fahrner[1], Donald Slater[2], Aman KC[3], Claudia Cenedese[4], David A. Sutherland[1], Ellyn Enderlin[3], Femke de Jong[5], Kristian K. Kjeldsen[6], Michael Wood[7,8], Peter Nienow[2], Sophie Nowicki[9], Till Wagner[10]

[1] University of Oregon, Department of Earth Sciences, USA
[2] University of Edinburgh, School of Geosciences, UK
[3] Boise State University, Department of Geosciences, USA
[4] Woods Hole Oceanographic Institution, Physical Oceanography Department, USA
[5] Royal Netherlands Institute for Sea Research, the Netherlands
[6] Geological Survey of Denmark and Greenland, Denmark
[7] Moss Landing Marine Laboratories, San José State University, USA
[8] Jet Propulsion Laboratory, California Institute of Technology, USA
[9] University at Buffalo, Department of Geology, USA
[10] University of Wisconsin-Madison, Department of Atmospheric and Oceanic Sciences, USA

*Correspondence to: Dominik Fahrner (dfahrner@uoregon.edu)*

## Abstract

**Frontal ablation at tidewater glaciers, which comprises iceberg calving, submarine and subaerial melting, is a key boundary condition for numerical ice sheet models but remains difficult to measure directly in-situ. Many previous studies have quantified frontal ablation over varying spatio-temporal scales, however most use ice discharge as an approximation for frontal ablation, thereby neglecting the influence of terminus location change. Frontal ablation estimates that do account for terminus location change are spatio-temporally limited by the availability of observational data. Here, we present a processing chain to quantify frontal ablation using open-source observational data. We apply the processing chain to 49 tidewater glaciers in Greenland with reliable near-terminus bathymetry data in the BedMachine V4 dataset. Near-terminus volume change over the time period 1987 - 2020 is determined using a previously published dataset of terminus positions (TermPicks), ice thicknesses from ArcticDEM and AeroDEM, adjusted for surface elevation change over time, and bathymetry data from BedMachine v4. Assuming a vertical terminus geometry and uniform ice density, we estimate frontal ablation as the difference between mass flux towards the terminus (Mankoff et al., 2020) and mass change between consecutive observation. The frontal ablation dataset offers exciting opportunities for developing**



**new insights into ice dynamics, including helping to improve numerical model**
**hindcasting and projections. Lastly, we provide a processing chain that may serve as**
**a community standard for determining frontal ablation from observational data for any**
**tidewater glacier.**

## Introduction

Greenland's tidewater glaciers have been accelerating and retreating since the mid-1990's
and contribute ~30-60 % of the total annual mass loss from the Greenland Ice Sheet (GrIS)
through frontal ablation (Enderlin et al., 2014; Mouginot et al., 2019; Shepherd et al., 2020).
Frontal ablation, which comprises iceberg calving, submarine melting and subaerial melting at
the glacier terminus, can be an important component of glacier mass balance and is
susceptible to changes over a wide range of time scales (e.g., through changes in ice flow,
ocean or air temperatures, or near terminus sea-ice or mélange conditions; e.g. Cowton et al.,
2018; King et al., 2020). The volume flux of ice across a fixed gate (referred to as discharge)
is often used to approximate frontal ablation, which does not take terminus position change
into account (Rignot and Kanagaratnam, 2006).
Studies that do determine frontal ablation while taking terminus position change into account
have been conducted over varying spatio-temporal scales (Osmanoğlu et al., 2013; McNabb
et al., 2015; Fried et al., 2018; Wagner et al., 2019; Kochtitzky et al., 2022, 2023) and using a
variety of data (Köhler et al., 2016; Wychen et al., 2020; Bunce et al., 2021). However, in situ
observational data, especially for the GrIS, are often lacking and satellite remote sensing data
are temporally limited by image availability. Multi-decadal estimates of frontal ablation are
therefore often confined to specific locations (e.g. McNabb et al., 2015) or limited time periods
(e.g. Köhler et al., 2016; Bunce et al., 2021). The most recent comprehensive study by
Kochtitzky et al. (2023) determined frontal ablation for all glaciers of the GrIS, however their
study is constrained to the use of decadal averages.



In current, large-scale, numerical ice sheet models, frontal ablation is heavily parameterized
and remains a key uncertainty for projecting future sea level rise (Luckman et al., 2015; Benn
et al., 2017; Slater et al., 2019; Goelzer et al., 2020). The limited understanding of frontal
ablation processes, partially due to the scarcity of observational data, and the lack of long
timeseries of frontal ablation further complicate the inclusion of ice-sheet-ocean processes in
numerical models (Cowton et al., 2018; Slater et al., 2019). Quantifying frontal ablation from
observational data is therefore crucial to improve our understanding of near-terminus ice
dynamics and improving numerical modelling efforts (Benn et al., 2017; Cowton et al., 2018)
The processing chain presented here derives multi-decadal time series of frontal ablation for
tidewater glaciers located along the Greenland coast using publicly available remote sensing
observational data. The high spatio-temporal resolution (up to monthly) of the resulting
timeseries can provide new insights into mass loss from tidewater glaciers in Greenland and
is aimed at improving the current understanding of ocean forcing of the GrIS.
**Product description**
At any tidewater glacier, there is a competition of processes that determine whether termini
advance, retreat or remain stable. The ice velocity at the terminus pushes the terminus
forwards, while calving and melting of subaerial portions of the ice face move the terminus
backwards (i.e., in the direction opposite to ice flow). This may be expressed mathematically
as

$$\int_A \frac{dL}{dt}\, dA = \int_A v\, dA - \int_A (c + m_s + m_a)\, dA \tag{1}$$


in which $L$ is terminus position, $v$ is ice velocity at the terminus, $c$ is calving rate, $m_s$ is
submarine melt rate and $m_a$ is subaerial melt rate. Each of these quantities may vary with
depth or width along the calving front but in Eq. 1 we integrate over the terminus frontal area
A. Note that we define $\frac{dL}{dt}$ as positive for glacier advance and negative for glacier retreat. We



define frontal ablation, $F$, as the sum of calving, submarine melting, and subaerial melting
rates; that is all the processes that remove ice from the calving front.

$$F = \rho_i \int_A (c + m_s + m_a)\, dA \qquad (2)$$

where the ice density $\rho_i$ is included so that $F$ is a mass flux. Quantifying frontal ablation directly
by estimating calving rate, submarine melt rate and subaerial melt rate is very difficult and
uncertain, but we can note from Eq. 1 that

$$F = \rho_i \int_A v\, dA - \rho_i \int_A \frac{dL}{dt}\, dA, \qquad (3)$$

which expresses frontal ablation in terms of frontal ice velocity and terminus position change.
If we assume that these are relatively depth-invariant (i.e., vertical terminus face and plug flow
of ice), then Eq. 3 can be rewritten as

$$F = \rho_i \int_W H v_s\, dW - \rho_i \int_W H \frac{dL_s}{dt}\, dW \qquad (4)$$

where $H$ is ice thickness, $W$ the width of the glacier, and $v_s$ and $L_s$ are the velocity and terminus
position at the glacier surface, estimated from readily available remote sensing datasets.
Hence, Eq. 4 provides a practical means of estimating frontal ablation. Note that the first term
on the right-hand side of Eq. 4 is commonly referred to as the solid ice discharge $D$ (e.g.
Mankoff et al., 2020) so that frontal ablation differs from solid ice discharge by the mass
change relating to terminus position change (hereafter referred to as *TMC*). We therefore
simplify Eq. 4 to

$$F = D - TMC. \qquad (5)$$

The data product presented here provides frontal ablation estimates for 49 selected tidewater
glaciers in Greenland using available terminus position observations from the TermPicks
dataset (Goliber and Black, 2021).
The tidewater glaciers included in the dataset were selected based on the reliability of methods
that were used to determine fjord bathymetry (Figure 1; Morlighem et al., 2017, 2021; Wood
et al., 2021). We include only glaciers where bathymetry data were derived from
measurements, mass conservation or the GIMP DEM (as classified in the BedMachine v4



dataset), thereby excluding glaciers where bathymetry was derived synthetically or by
interpolation, kriging, or gravity inversion (Morlighem et al., 2017, 2021). However, the
presented workflow can be applied to any glacier, independent of the reliability of bathymetry
data, provided that the data outlined in the *Data Sources section* are available.

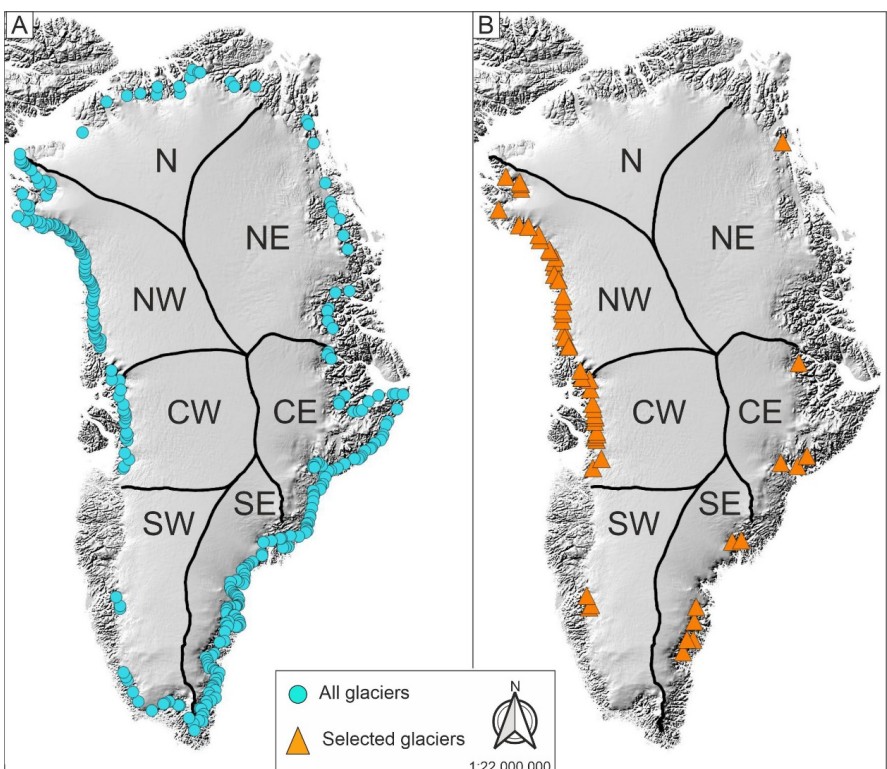

**Figure 1 | Overview of Selected tidewater glaciers**

A) Location of all tidewater glaciers for which terminus observations are available in the TermPicks dataset (Goliber and Black, 2021). B) Location of tidewater glaciers that have been selected for this study based on the reliability of bathymetry data. The basemap is taken from BedMachine v4 (Morlighem et al., 2017, 2021); lines show drainage basins after Mouginot and Rignot (2019).







## Data sources


This section introduces previously published data sources, which are publicly available and were used to calculate frontal ablation. The spatio-temporal resolution of each input dataset as well as the associated uncertainties can be found in Tables S1 and S2.

### Terminus positions

We use terminus position data from the TermPicks dataset (Goliber and Black, 2021), which includes manually as well as automatically delineated terminus positions from various sources for the period 1916 to 2020. The availability of terminus delineations varies over time as satellite imagery prior to the start of NASA's Landsat program in 1972 is sparse. The TermPicks dataset and its metadata are considered standardized. However, due to the different sources of the individual terminus delineations, additional filtering is required before they can be used to quantify terminus change over time (see Methods).

### Surface elevation, bathymetry, and ice thickness

We use glacier specific surface elevation change rates determined by Khan (2017) for the period 1995 – 2015. These are combined with the latest ArcticDEM image that covers the full extent of the tidewater glacier at its most advanced and most retreated position (Porter et al., 2018) and AeroDEM, which has been derived from stereophotogrammetric imagery recorded during 1978-87 (Korsgaard et al., 2016b). Bedrock topography is taken from BedMachine v4 (Morlighem et al., 2017, 2021). Time series of ice thickness are computed using the DEMs, adjusted to account for surface elevation change, and bedrock topography (see Methods).

### Ice Velocity

Ice velocities are taken from NASA's Making Earth System Data Records for Use in Research Environments (MEaSUREs) Inter-Mission Time Series of Land Ice Velocity and Elevation (ITS_LIVE) project. We use composite images, which provide annual flow velocities for the period 1985 – 2018 (Gardner et al., 2019).



**Discharge**
Solid ice discharge data with uncertainties is taken from Mankoff et al. (2020) for the period
1986 – 2020 and is used to calculate frontal ablation as shown in Eq.5. The flux gates used
to derive solid ice discharge are located approximately 5 km upstream of the terminus, so that
there could be a time lag and/or difference between the solid ice discharge estimated at the
flux gate the discharge at the terminus. Mankoff et al. (2020) estimate the difference in
discharge between gates located 1 km and 5 km from the terminus to be around 5% at the ice
sheet scale, but it is unclear how much of this difference arises due to uncertainty in bed
topography, which generally increases closer to the terminus. Acknowledging this small
possible difference, together with the strong longitudinal stress coupling at fast-flowing
tidewater glaciers (e.g. Enderlin et al., 2016), we here take the discharge from the flux gates
5 km upstream to be representative of the flux at the terminus. Following Mankoff et al. (2020),
we also assign an error of ~10% to these discharge values (see later).
**Satellite Imagery**
NASA Landsat 8 satellite imagery are downloaded from NASA's Earth Explorer for each
individual glacier, and true-color, panchromatically sharpened images are created using the
red, blue, green, and panchromatic band (Bands 2, 3, 4 and 8). The pan-sharpened true-color
images are used to manually digitize fjord walls, which are used to bring terminus delineations
to a consistent length and to create polygons for each observation.
# Methods
**Fjord geometry**
Tidewater glacier terminus positions in the TermPicks dataset vary widely in their length and
differ in their starting/end points. For example, traces for an individual glacier could be drawn
in opposing directions (e.g., from North to South as well as South to North), which will be
referred to as drawing direction hereafter, depending on the author. This variability in terminus
trace drawing direction therefore necessitates standardization for further processing.



Here fjord boundaries are created by manually delineating the upper and lower fjord walls
using pansharpened NASA Landsat 8 imagery, with the coordinates of the boundaries being
saved so that this step only has to be completed once. Subsequently, the drawing direction is
standardized based on the distance between the terminus delineation endpoint and the fjord
walls. If the end point of the terminus delineation is located closer to the lower fjord wall than
the upper fjord wall the terminus delineation is rotated (Figure 2).

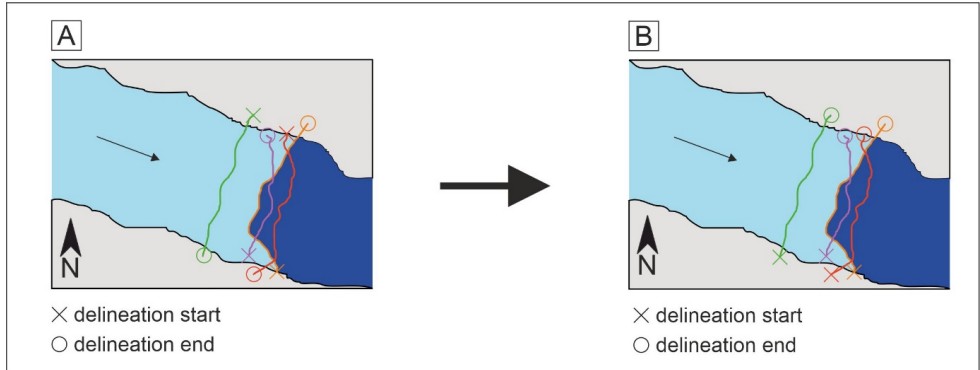

**Figure 2 | Sketch of drawing direction**

A) Example of drawing directions of terminus delineations in TermPicks dataset, with termini start and endpoints (indicated by *x* and *o* symbols respectively) indicating if they have been traced North to South or South to North. B) Terminus delineations after standardizing drawing directions for further processing.


## Definition of upstream boundary

A reference boundary needs to be defined so that individual terminus positions can be
compared to each other. This boundary is defined manually by drawing an arbitrary line
upstream of the most retreated position of the glacier that intersects both fjord walls on top of
a pansharpened true color NASA Landsat 8 image. The reference boundary is fixed and
remains the same for all terminus positions at a given glacier over time.
Polygons are created for each terminus observation by combining the reference boundary,
respective terminus delineation and the fjord wall boundaries between these two locations.
The polygons, herein referred to as the area of interest (AOI), provide the basis for the
calculation of area and volume change between observations.

## Terminus positions

The TermPicks delineations of all investigated tidewater glaciers are first visually examined to identify obvious outliers caused by e.g., false georeferencing of the satellite image or delineation of mélange. We then subsampled the TermPicks dataset (Goliber and Black, 2021) by using approximately monthly terminus traces selected as the closest in time to the 1st of the month in each month. We also restricted the terminus positions to lie within the period of ice discharge estimates (1986-present; Mankoff et al., 2020). While this reduces the amount of terminus position observations, at times drastically, we found that the uncertainties associated with delineations created by different authors are too high to ensure an accurate product at higher temporal resolution. We chose delineations based on their time difference to the 1$^{st}$ day of the respective month to enable subsequent temporal averaging.

In a second step, the monthly terminus observations are filtered to remove erroneous delineations (e.g., due to false geolocation of the underlying satellite image) and to ensure consistency in the dataset. The filtering is conducted in multiple, sequential steps, as follows:

1) Terminus delineations are removed if they contain more than one line segment, which can occur, for example, if the terminus is split by a nunatak and the terminus has been delineated in two parts. While it would be possible to linearly interpolate between the line segments, this would skew the data and introduce unnecessary errors when combined with fully-delineated termini.

2) Delineations which are smaller than 95 % of the terminus width, which is defined as the minimum distance between the two fjord walls, are excluded from further analysis. We further exclude terminus positions that are longer than the mean fjord width plus two standard deviations of the mean terminus length, as theses delineations would skew the subsequent mass change calculations (Figure 3). This filtering step ensures that the delineations used for further analysis represent the glacier terminus accurately.



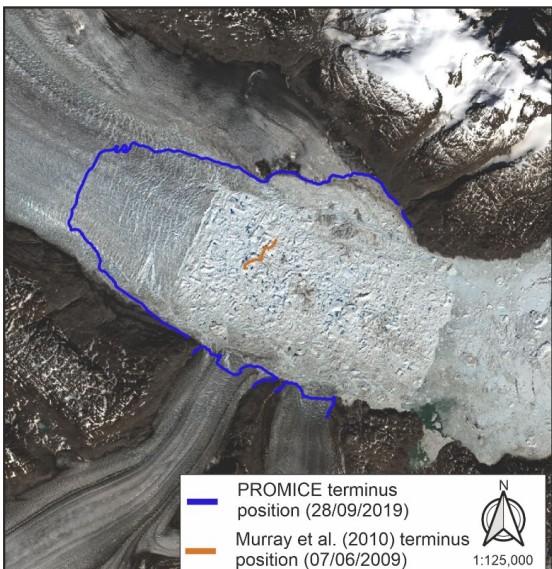

**Figure 3 | Example of unusable delineations**

The figure shows a terminus delineation from 28/9/2019 by PROMICE (blue) and from Murray et al. (2010) from 07/06/2009 (orange) for Kangerlussuaq Glacier on top of a panchromatic RGB (band-stacked) NASA Landsat 8 image from 30/6/2021. These delineations are examples of unusable terminus traces as they are significantly shorter than the actual terminus width (Murray et al., 2015) or include delineations of the fjord walls (PROMICE).


3) We use NASA MEaSUREs ice velocity to filter out terminus traces that indicate the
front has advanced faster than the ice velocity, which is physically not possible. The
annual composite velocity images are automatically downloaded when first running the
code (Greene et al., 2017). Ice flow velocities are subsequently extracted along a
centerline between the most retreated and most advanced terminus position for each
glacier. This method is chosen to ensure that velocities are representative of the
terminus region and are not skewed by slower flowing parts of the glacier (e.g., lateral
drag at the margins). The flow velocities are then averaged for the decade preceding
the last available velocity observation to create a decadal mean value velocity.
Terminus advance is determined using the normal $n$ to the connection of midpoints ($m_1$
and $m_2$) of subsequent delineations ($t_1$ and $t_2$; Figure 4). If the midpoint of the
subsequent delineation is located down-fjord of the normal, the glacier movement is
classified as advance. Terminus advance or retreat is quantified by calculating the
distance between delineation midpoints along a centerline. Then, to ensure that the
delineations represent realistic changes in terminus positions, we use the decadal
mean near-terminus velocity to infer how much the glacier could have advanced during



each terminus observation timestep. If the terminus advance is greater than twice the
predicted flow velocity, the terminus delineation is considered erroneous and is
excluded from the dataset.

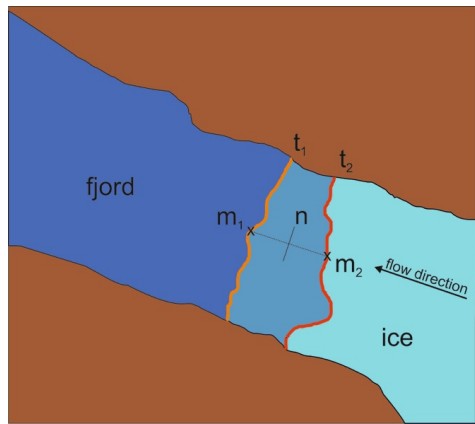

**Figure 4 | Sketch of glacier advance/retreat determination for filtering**

The normal (n) to the line connecting the midpoints ($m_1$ and $m_2$) of subsequent terminus delineations ($t_1$ and $t_2$) is used to determine advance and retreat of the glacier. In the sketch shown here, $m_2$ is right of the normal n, therefore the glacier has retreated. The classification of advance/retreat is solely used for filtering purposes.

During this phase we further exclude the originally selected glaciers Kjer and Nordenskiöld
from the analysis as the fjord walls are extremely difficult to delineate, as well as Zachariae
Isstrøm and Qeqertaarsuusarsuup Sermia due to their floating ice shelves to reach the final
49 selected systems. The manual delineation of ice shelves is challenging due to their complex
structure and the difficulty of distinguishing between terminus and mélange. It should be noted
that input terminus delineations should be as accurate as possible to avoid large uncertainties
in the frontal ablation estimates. Overall, after quality control and temporal filtering, the dataset
contains 34.9 % of all terminus delineations (6674 of 19120; Table S1) for the selected 49
glaciers. An overview of the number of terminus positions pre- and post-filtering can be found
in Figure S1. The filtered dataset is the basis of all further analysis and is written to individual
shapefiles for manual quality control and to speed up future processing.
Subsequently, in order to accurately compute terminus area change using the filtered time
series, the length of each terminus position must be set to be consistent with the previously
defined fjord wall boundaries. To determine the location of the start/end points in relation to
the fjord walls, the fjord wall polylines are converted into a polygon. If the start/end point lies



within the respective boundary polygon, the terminus delineation is clipped, and the new
start/endpoint is defined as the intersection point between terminus delineation and boundary
polygon (Figure 5 A, B). If the point lies outside of the boundary polygon, the terminus
delineation is extrapolated to the nearest point on the boundary polygon (Figure 5 C, D). We
compare the length of the extrapolation to the length of the manually delineated terminus trace
to ensure that the majority of the terminus is captured by the latter. The observation is excluded
if the length of the extrapolation exceeds the length of the delineated terminus.
These processing steps are conducted for the upper and lower boundary separately and
terminus delineations are saved once completed.

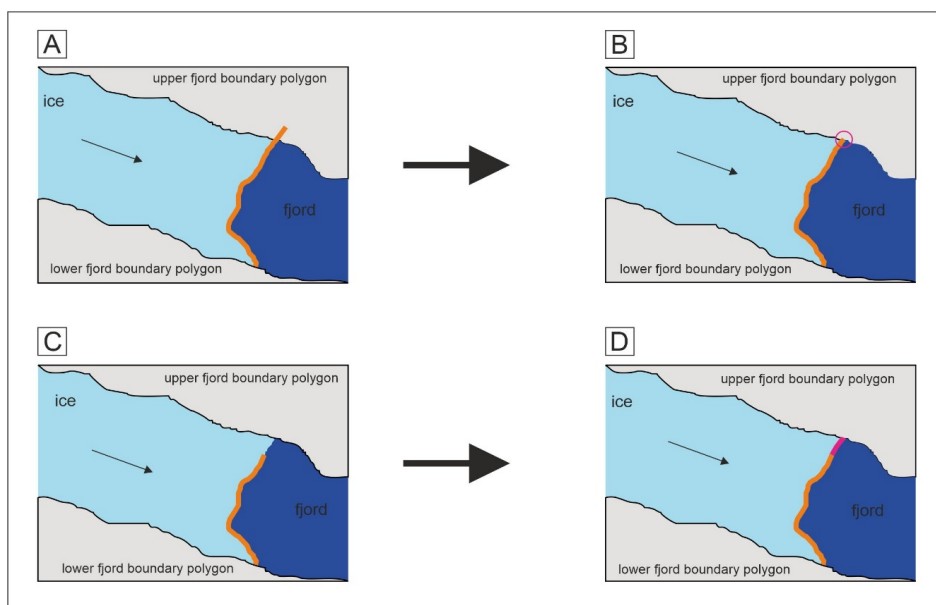

**Figure 5 | Sketch of trace cropping/extrapolation**

Sketch showing how terminus delineations are cropped or extrapolated. A) If terminus delineation is drawn
across the fjord boundary, terminus delineation is cropped to the intersection with the fjord boundary (B). C)
If terminus delineation does not intersect or reach the fjord boundary, the delineation is extrapolated to the
nearest point on the fjord boundary from the delineation endpoint (D). Small arrow shows glacier flow direction.

**Surface elevation change and ice thickness**
Ice surface elevation is estimated for the terminus area at the time of each individual terminus
observation based on i) Khan (2017) if surface elevation change data are available for the



individual tidewater glacier (hereafter referred to as Khan surface change rate or *K-SCR*), and
ii) the elevation difference between the ArcticDEM (Porter et al. 2018) and the AeroDEM
(Korsgaard et al., 2016a) divided by the time difference (hereafter referred to as ArcticDEM-
AeroDEM surface change rate or AA-*SCR*). A workflow schematic is shown in Figure 6.
If K-SCR data are available and the terminus observation date (TOD) is within the K-SCR time
range, the annual mean of the K-SCR within the AOI is calculated. The elevation of the latest
ArcticDEM is then adjusted by summing the K-SCR for the time difference between the
ArcticDEM and terminus observation date (Figure 6). For terminus observations outside the
K-SCR time range, the elevation of the ArticDEM is adjusted by summing the K-SCR for all
available dates and adding the AA-SCR multiplied by the time difference between the end of

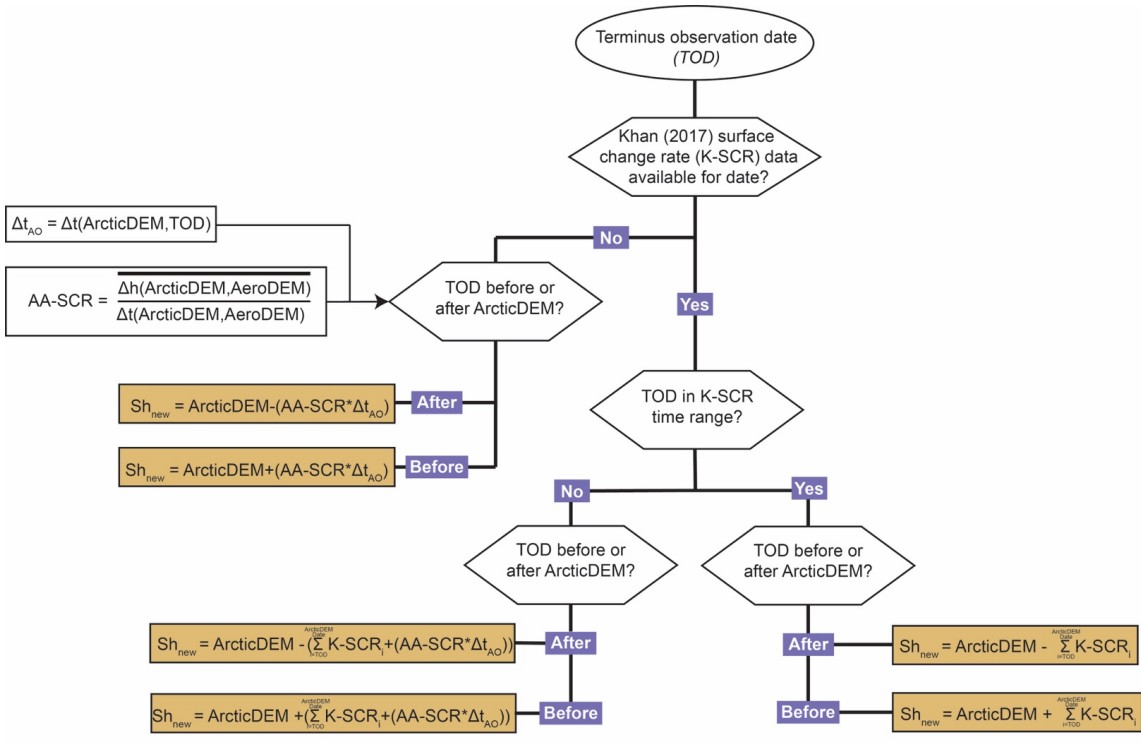

**Figure 6 | Process chart for determining surface elevation**

The process chart shows how surface elevation for a given terminus observation is determined based on
the date of the terminus observation (TOD), the availability of surface elevation change rate data from
Khan (2017) referred to as K-SCR, and the difference between ArcticDEM and AeroDEM divided by their
time difference (AA-SCR).





K-SCR data and the terminus observation. This allows one to account for surface elevation
change prior/after the K-SCR time range, assuming that the surface elevation change is linear
for that time period (Figure 6).
If no data are available from Khan (2017) for the selected tidewater glacier, the SCR is
multiplied by the time difference between terminus observation and the ArcticDEM. The
resulting surface change rate is then subtracted from or added to the ArcticDEM based on the
date of the terminus observation (added if the terminus observation is earlier than the
ArcticDEM). This method assumes linear surface elevation change and does not account for
intra- or inter-annual variability, which introduces uncertainties that could influence the frontal
ablation calculation.
The ice thickness H for each terminus observation is then calculated as the difference between
the adjusted surface elevation from the underlying bedrock topography (Morlighem et al.,
2017, 2021). In the subsequent processing step, the volume for each individual terminus
observation polygon is calculated by multiplying the area $A_P$ of the polygon with the mean ice
thickness H.
**Frontal Ablation Calculation**
We first calculate the mass for each terminus observation using the previously created 3-D
polygon and an ice density of $0.917 \text{g/cm}^3$ (Figure 7). The presence of significant crevassing
near the termini of tidewater glaciers means that some portion of the polygon is in fact air
rather than ice, so that the effective density of the polygon will be smaller than that of pure ice.
We are not aware of a study that estimates such an effective density, but on the basis of
papers that have mapped crevasses (e.g. Enderlin and Bartholomaus, 2020; Van Wyk de
Vries et al., 2023) we do not expect a substantial difference from the density of pure ice. As
such, we proceed with the pure ice density, but we bear in mind that this may be an upper
bound. For each timestep, the respective mass is then linearly interpolated to the first of each
month, and mass change over the month in Gt/d is then calculated as the difference in the



mass divided by the number of days in the month. The same processing steps are then applied
to ice discharge (D) and finally, frontal ablation (F) is calculated as in Eq. 5. With the applied
interpolation and averaging, the results can be interpreted as the mean value over the month
in question. The final dataset of frontal ablation contains the interpolated as well as the original
values of mass change and solid ice discharge. A sketch of the mass change calculation is
illustrated in Figure 7.
The processing chain provides the possibility to estimate monthly, three-monthly or annual
frontal ablation. Note that the frontal ablation estimate should be considered as an average
over the time period. We recommend use of the three-monthly or annual estimates because
the monthly estimates are more susceptible to errors in the terminus delineation induced for
instance by pixel size of the satellite image or individual delineation error. In the results shown
below we present the three-monthly estimates.

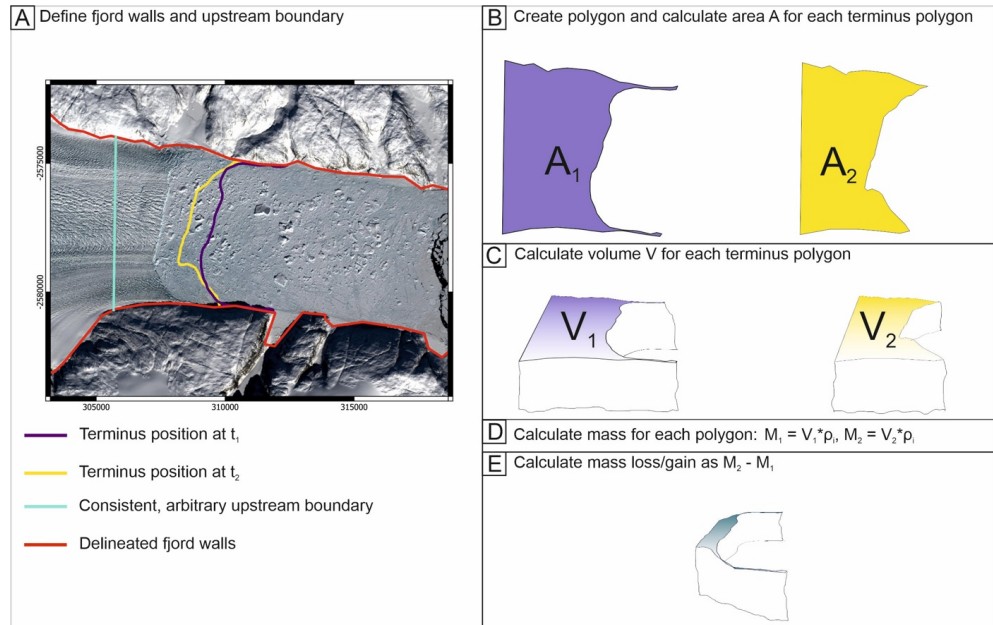

**Figure 7 | Sketch of mass loss calculation**

A) Example of two consecutive terminus delineations (purple at $t_1$, yellow at $t_2$), fjord wall boundaries (red) and upstream boundary (cyan) at Helheim Glacier, SE Greenland, on top of panchromatically sharpened NASA Landsat 9 image from 06/04/2022. B) Created polygons with area $A_1$ and $A_2$, corresponding to terminus delineations at times $t_1$ and $t_2$. C) Created 3D polygons with volume $V_1$ and $V_2$, corresponding to terminus delineations at $t_1$ and $t_2$ D) Calculate Mass $M_1$ and $M_2$ using $M=V\rho_i$. E) Determine mass change between terminus delineations defined as difference between masses $M_1$ and $M_2$ (which in the case shown here is negative).



## Uncertainty quantification


The above-described input data products contain glacier- and time-dependent uncertainties,
so that errors are introduced to the frontal ablation estimates presented here. To quantify the
uncertainty in frontal ablation estimates for each individual tidewater glacier investigated in
this study, the uncertainties of the input data products are propagated through the processing
chain using error propagation.
Frontal ablation is defined as the difference between solid ice discharge and mass change at
the terminus (Eq. 5; Cogley et al., 2011). If the error on solid ice discharge (D) is $\Delta D$, which
we take from Mankoff et al. (2020), and the error on terminus mass change (TMC) is $\Delta TMC$,
then the uncertainty in frontal ablation $\Delta F$ is:

$$\Delta F = \sqrt{(\Delta D)^2 + (\Delta TMC)^2} \tag{6}$$

We neglect the uncertainty of ice density ($\rho_i$; cf. Mankoff et al. (2020)) and calculate TMC as
the difference between two volumes separated by a time $t_2 - t_1$ (Figure 7):

$$TMC = \rho_i \frac{V_2 - V_1}{t_2 - t_1} \tag{7}$$

To estimate the error on TMC, we neglect uncertainty in the ice density and approximate the
difference between the volumes as a cuboid of width W, thickness H and length L (i.e., if the
glacier has retreated between $t_1$ and $t_2$ then W is the fjord width, H is the ice thickness and L
is the retreat length). Neglecting the error on fjord width, we can then estimate the error on $V_2$-
$V_1$ as

$$\Delta V = \sqrt{W^2 H^2 \Delta L^2 + W^2 L^2 \Delta H^2} \tag{8}$$

where $\Delta H$ = maximum ice thickness error and $\Delta L$ = terminus delineation error.
The delineation uncertainty $\Delta L$ is based on the satellite that was used to delineate the terminus
position. While previous studies suggest relatively small delineation errors, these estimates
are for a single operator and only for Landsat 7/8 and Sentinel 1 (Brough et al., 2019; Fahrner

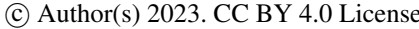

et al., 2021). To account for multiple operators and varying satellites, we choose to keep the
delineation error constant at 30 meters, which is the average pixel resolution of Landsat
satellites (Landsat 4–6-pixel resolution: 60 meters; Landsat 7–8-pixel resolution: 15 meters).
The maximum terminus change is time-averaged over a user defined period (monthly, three-
monthly, annually) to conform with the calculation of frontal ablation. Combining Eqs. 7 and 8,
we can then determine the uncertainty (ΔTMC) on terminus mass change for each time-
averaged step as:

$$\Delta \text{TMC} = \frac{\rho_i}{t_2 - t_1} \Delta V \qquad (9)$$

Where $t_2$-$t_1$ is the time resolution of the frontal ablation dataset (31 days, 90 days, 365 days),
chosen as 90 days in our results. Eq. 9 gives uncertainties which change in time, but for
simplicity in the results and analysis we take a single value which is the maximum over the
analysis period. With the described uncertainty of terminus mass change and discharge
estimates (Mankoff et al., 2020), we can ultimately calculate the maximum uncertainty of
frontal ablation using Eq. 6.

**Results**
Results for all investigated tidewater glaciers with observation-based bed geometries can be
found in Figures S2 – 53. As an example of the impact of terminus change on frontal ablation
time series, the frontal ablation and solid ice discharge time series for Helheim Glacier, SE
Greenland are shown for the period 1987 – 2020 (Figure 8). The temporal resolution and
coverage of the data shown for Helheim Glacier is representative of all study sites.
In accordance with an increase in ice velocity and terminus retreat, frontal ablation for Helheim
glacier in SE Greenland shows a sharp rise starting in 2004/05 (Figure 8A-C). These results
are consistent with a large-scale retreat during this time frame as determined by previous



studies (e.g. Howat et al., 2005, 2008). The results further show that relatively high frontal
ablation rates remain present over the following decade and are accompanied by sustained
yet seasonally varying terminus retreat, decreasing ice velocities and relatively stable ice
discharge (Figure 8B - D). To highlight the influence of terminus position change on frontal
ablation, colors shown in Figure 8D correspond directly to the terminus positions used in
calculating frontal ablation (Figure 8E). It is seen that while the ice discharge has limited
seasonal variability (Fig. 8C), periods where there is seasonal advance and retreat of the
terminus (Fig. 8B) result in seasonal variability in frontal ablation (Fig. 8C). The sustained
period of retreat from 2000-2005 is driven by frontal ablation values that frequently exceed the
ice discharge and reach up to 50% above the ice discharge for three-month periods. It is
apparent from Figure 8C that frontal ablation estimates that take terminus change into account
show a higher variability than those derived from ice discharge alone (Mankoff et al., 2020).

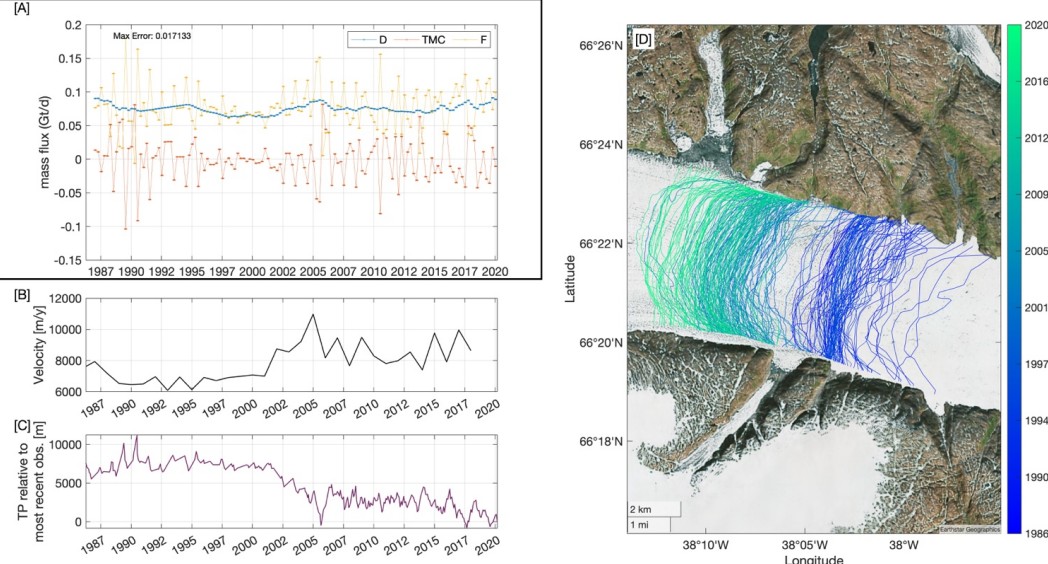

**Figure 8 | Example of output data**

Example of output data for Helheim Glacier, SE Greenland shown in A) with three-monthly frontal ablation estimates shown in yellow, and discharge (blue) and terminus associated mass change (TMC, red) shown for comparison. Maximum error is also shown (for details see supplementary). B) Annual flow velocity in m/yr from NASA ITS_LIVE data B) Terminus position (TP) relative to most recent observation along the centerline. Panels B) and C) are only shown for validation purposes and are not part of the dataset. D) Terminus positions used to calculate frontal ablation estimates colour coded by date.



Figure 9 shows the annual average frontal ablation for a period where terminus observations
are available for all tidewater glaciers (1987 – 2018). Helheim Glacier, Kangerlussuaq Glacier
and Sermeq Kujalleq (Jakobshavn Isbræ) contribute the most to the total frontal ablation of
the investigated glaciers (Figure 9, Table S2). However, seven additional tidewater glaciers
around the GrIS show comparatively large frontal ablation values for the same time period
(namely: Kangiata Nunaata Sermia (5.65 Gt/yr), Nansen Glacier (6.35 Gt/yr), Sermeq Kujalleq
in the Central West (7.68 Gt/yr), Sermeq Kujalleq (Store Glacier; 9.14 Gt/yr), Daugaard-
Jensen Glacier (9.48 Gt/yr), Kangiliup Sermia (Rink Isbræ; 12.78 Gt/yr), and Tuttulikassaap
Sermia (13.35 Gt/yr). The majority of the studied tidewater glaciers (31 glaciers or ~63 %)

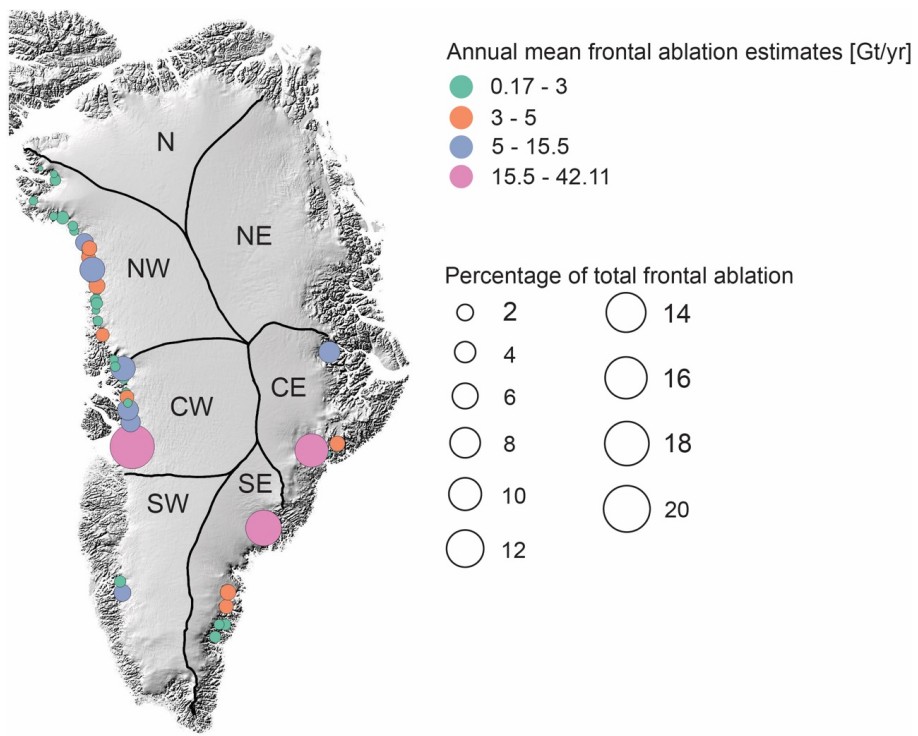

**Figure 9 | Overview of total frontal ablation 1987-2015**

Overview of total frontal ablation (sum of three-monthly averages) for all tidewater glaciers investigated in this study for the period 1987-2015 (period when terminus observations are available for all tidewater glaciers). Colors show the mean annual frontal ablation estimate for each tidewater glacier; Circle size indicates contribution of mean frontal ablation to the total frontal ablation of all tidewater glaciers. The basemap is taken from BedMachine v4 (Morlighem et al., 2017, 2021), with lines indicating drainage basins (Mouginot et al., 2019)



have frontal ablation values smaller than 3 Gt/yr while eight glaciers have mean annual frontal
ablation values between 3 and 5 Gt/yr (Table S2).
The processing chain presented here provides a novel way to estimate frontal ablation rates
over long temporal scales, while also taking changes in terminus position into account. The
results show that over seasonal timescales (e.g. 3 months), frontal ablation rates that take
terminus position change into account can be significantly higher than estimates from ice
discharge alone, thereby highlighting the importance of including terminus variability at these
timescales. A recent study calculated decadal mean frontal ablation estimates for essentially
all glaciers in Greenland for the period 2000-2010 and 2010-2020 (Kochtitzky et al., 2023). In
comparison our study focuses on fewer glaciers but at higher temporal resolution. When
comparing glaciers that are included in both studies, and calculating decadal mean values
from our dataset, we find that for the period 2000-2010 the majority of our estimates (>80%)
are within the uncertainty boundaries of Kochtitzky et al. (2023; Figure 10; Table S 3).
Agreement is reduced for the period 2010-2020, with roughly half of our frontal ablation
estimates (~51%) agreeing with Kochtitzky et al. (2023) within uncertainty, however we find
higher estimates for ~40% (17 glaciers) and lower estimates for ~9% (4 glaciers). For all
tidewater glaciers investigated in this study, we estimate total decadal frontal ablation to be
217.1 ± 68.6 Gt/yr for the period 2000-2010, and 245.5 ± 68.6 Gt/yr for the period 2010-2020,
which agrees within uncertainty with the total over same glaciers in Kochtitzky et al. (2023;
Table S3). It should be noted that our study has a very different temporal resolution to
Kochtitzky et al. (2023) – three-monthly here versus decadal in their study – and that when
summing our three-monthly values to obtain decadal means we cautiously assumed that the
errors are fully systematic, which explains the larger uncertainty bounds on our decadal values
quoted above.

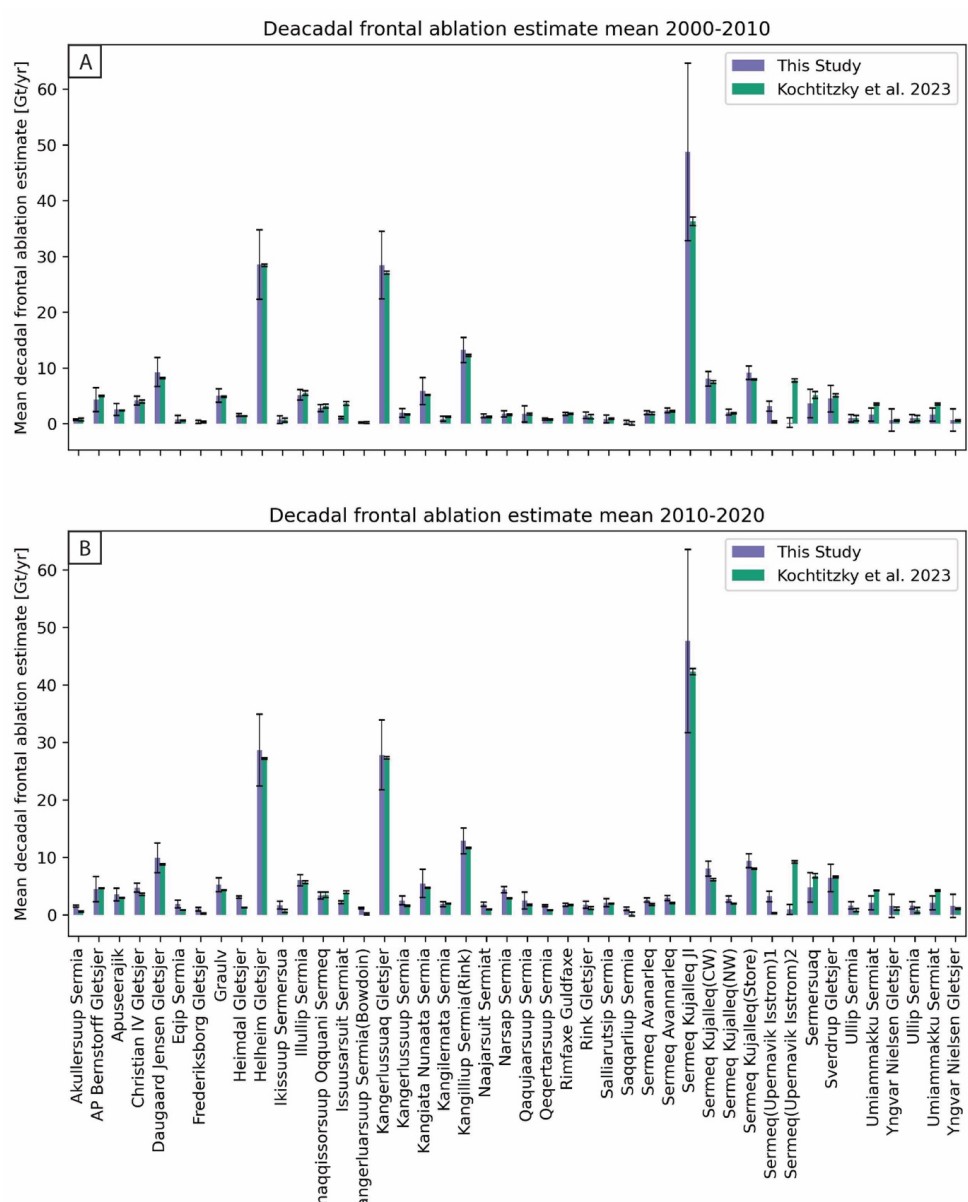

**Figure 10 | Comparison of results from Kochtitzky et al. (2023) and this study**

A) Comparison of decadal mean frontal ablation estimates presented in this study to results from Kochtitzky et al. (2023) for the period 2000-2010 with uncertainties. B) Comparison of decadal mean frontal ablation estimates presented in this study to results from Kochtitzky et al. (2023) for the period 2010-2020 with uncertainties.




The differences in decadal frontal ablation values between our study and Kochtitzky et al.
(2023) may result from the higher temporal resolution of terminus delineations used in this
study; values which have to be summed to get decadal means in order to do the comparison.
However, a degree of the difference also arises from the ice discharge – for example at
Sermeq Kujalleq (Jakobshavn Isbræ), the 2000-2010 mean ice discharge used in this study
is 41.8 Gt/yr (Mankoff et al., 2020) compared to 34.0 Gt/yr in Kochtitzky et al. (2023).
This study aims to provide the basis for further investigation of the influence of ocean forcing
on tidewater glacier termini, and to enable the generation of improved parameterizations of
ocean forcing for numerical ice sheet models. Importantly, the processing chain can easily be
modified to apply to any tidewater glacier, provided that the following data are available:
1) Terminus positions bracketing the time period of interest.
To determine frontal ablation on inter-annual or smaller timescales, we suggest
providing as many terminus delineations that have been created by a single operator
as possible to reduce spatial variability between observations.
2) A satellite image for manual delineation of fjord and upstream boundaries
3) Bedrock topography and surface elevation data to calculate ice thickness for the
individual terminus positions.

## 412 Conclusion

The dataset presented here provides three-monthly frontal ablation estimates for 49 tidewater
glaciers based on terminus position changes for each glacier, yet the processing chain can
easily be used to investigate frontal ablation at different temporal resolutions for example
monthly or annual. The dataset offers opportunities for the community to investigate the drivers
of mass loss at tidewater glaciers in Greenland and provides the basis to improve current
parameterizations of climate forcing in model hindcasting and projections.
The results show that over seasonal or shorter time periods, formal ablation estimates that
include terminus position change can differ significantly (up to ~50%) from those that are



derived from ice discharge alone. This illuminates the seasonal variability in frontal ablation at
tidewater glaciers and may shed light on the processes that drive mass loss at tidewater
glacier termini.
A brief example analysis of the dataset shows, that Sermeq Kujalleq (Jakobshavn Isbræ),
Kangerlussuaq glacier and Helheim glacier dominate annual frontal ablation estimates in
Greenland. However, we also find eight tidewater glaciers with comparatively high frontal
ablation estimates, which highlights that large mass loss is not necessarily confined to the
most dynamic tidewater glaciers. The results presented here are in agreement with a
previously published dataset (Kochtitzky et al., 2023) when considering the sum of frontal
ablation over all 49 glaciers, but differences exist at individual glaciers. However, the focus of
this study is on fewer glaciers at seasonal time resolution, making the dataset suitable for
investigating terminus mass loss processes.
We also hope that the processing chain will be a useful tool to quantify frontal ablation for any
glacier, as it is computationally inexpensive and can be adjusted easily. We further plan to
develop the processing chain into a standalone tool that can be hosted on GHub, thereby
making it fully open source to the community.
**Glossary**
*Solid Ice discharge [Gt/yr]:* Volume of ice flowing through a defined transect (or gate) upstream
of the terminus.
*Ice velocity [m/yr]:* Flow velocity of the ice as determined from NASA Landsat satellite imagery.
*Surface elevation change rate [m/yr]:* Change in surface elevation over time.
*Terminus Mass change [km³/timestep]:* Change of mass, which is calculated as near-terminus
volume change times ice density (917 kg/m³).
*Frontal ablation [Gt/d]:* Total loss of ice at the glacier front, which comprises iceberg calving,
submarine melting and subaerial melting (Truffer and Motyka, 2016). We define frontal
ablation as the difference between terminus mass change and solid ice discharge, with the
sign of mass change being dependent on the terminus configuration (advance = positive,
stable =0, retreat=negative).



**Author Contributions**


DF and all co-authors conceived the study. DF created the processing chain, pre-processed
the data, conducted all data analysis, figure production, and led the manuscript writing. All co-
authors provided conceptual and technical advice.

**Competing financial interests**


The authors declare no competing financial interests.

**Corresponding Author**


Correspondence and request for material should be addressed to dfahrner@uoregon.edu

**Data Availability**


The complete dataset with uncertainties can be found at
https://zenodo.org/records/10278419(Fahrner et al., 2023a).
Greenland Ice Sheet drainage basins can be found at https://doi.org/10.7280/D1WT11
(Mouginot and Rignot, 2019). Tidewater glacier terminus positions from the TermPicks
dataset can be found at https://zenodo.org/records/5117931 (Goliber and Black, 2021).
ArcticDEM data can be accessed at https://doi.org/10.7910/DVN/OHHUKH (Porter et al.,
2018). AeroDEM data can be accessed at https://doi.org/10.7289/v56q1v72 (Korsgaard et
al., 2016a). Bedmachine v4 bedrock topography data can be accessed at
https://doi.org/10.5067/VLJ5YXKCNGXO (Morlighem et al., 2021). Solid Ice discharge data
can be found at https://doi.org/10.22008/promice/data/ice_discharge/d/v02 (Mankoff et al.,
2020). Surface elevation change rates can be found at
https://doi.org/10.22008/FK2/GQJJEA (Khan, 2017).
The processing chain to produce this dataset including example data and a tutorial is available
at https://zenodo.org/records/10278429 (Fahrner et al., 2023b). The processing chain will also
be made available on GHub, so that it can be run as a standalone tool.

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

temporal-variability-at-narsap-sermia-
greenland/0B88B733EEB4C04C8C7AA9466FCB99A6.