# Peer review of "A Frontal Ablation Dataset for 49 Tidewater Glaciers in 1"

_Earth System Science Data, 2023_

## Referee Comment (RC1)

**General comments**

This manuscript describes the method used to generate a new dataset of frontal ablation for a sample of Greenland's tidewater glaciers. The dataset and workflow will be a great asset to the community and I am confident that the products will enable lots of good science. Overall, the method appears sound but I think the pre-processing of the terminus delineations and the estimate of ice thickness changes need to be improved to avoid erroneous changes in glacier volume. The majority of the manuscript is used to describe the method for calculating mass changes owing to terminus advance and retreat – I think more attention needs to be devoted to describing and documenting the dataset itself, and to quantifying the impact of various choices through the processing chain. In particular, the contribution of advance/retreat to frontal ablation should be quantified for all glaciers, individually and combined, at a range of temporal scales. I think the error metric for terminus mass change needs to be improved, particularly through the inclusion of thickness errors.

**Specific comments**

Data product:

The errors in D and TMC should also be provided, so that users can see how they contribute to the error in F.

I think it would be helpful if the time-series for each glacier were combined into a single file, perhaps a NetCDF. This would make it much easier for potential users to amalgamate the measurements from different glaciers.

A few issues that I assume are minor:

- For Akullersuup Sermia, the last two entries for TMC and F are empty, but D and Max_Error_F are provided. Is that expected?
- Time period: Again for Akullersuup Sermia, data are available for the period 1988 to 2018. In the manuscript, the abstract states 1987-2020, the caption of Figure 9 states 1987-2015 but line 361 of the submitted version states 1987-2018. Do different glaciers have different temporal coverage, with some (a few or most?) extending to 2020? If the glaciers do have different time periods, it would be easier for users if you at least make all of the time-series the same length by padding with NaNs as necessary.

Data sources:

Initially, I found this section a little confusing. It reads somewhat like an abstract for the methods, giving a very brief overview but without much detail, some of which is repeated again in the methods. The purpose of this section is just to tell the reader what data you used – giving them some identifier or name – so that you can refer to them unambiguously in your methods. With that in mind, I wonder if this section might be better structured as a series of bullet points where you clearly state the parameter (terminus position, elevation, ice velocity, discharge) and what data source(s) were used, then bring Supplementary Table into the main text at this point. I would recommend adding the time-period to the table (and possibly to the text describing the data).

Methods – fjord geometry and terminus positions

Line 169: lower/upper fjord wall is hard to understand. Should it be southern/northern? If so, then is the code able to deal with glaciers flowing north to south or south to north? Could you use true left or right instead – that would be independent of glacier orientation, but it would entail bringing in the flow direction data earlier, so might not be worth the additional effort.

Fjord wall delineation and polygon creation: overall, this approach for creating polygons seems very robust, but I have a few queries and have spotted a few areas where the algorithm appears to struggle:

1. For ice streams like Jakobshavn Isbrae/Sermeq Kujalleq, where there is no clear lateral topographic boundary in the vicinity of the present-day terminus, how did you determine where to place the fjord walls?

2. As glaciers thin, they can reduce in width. Depending on the date of the image used to digitise your fjord walls, you might be slightly (and it probably is only very slightly) overestimating or underestimating the change in area owing to terminus position change in years far before or after the image was acquired. I think it would probably be more effort than it's worth to quantify the impact of this because the associated error is likely smaller than other errors, but I suggest that this effect is acknowledged somewhere.

3. Definition of terminus width when filtering polygons: As I understand it, the purpose of this step is avoid making long extrapolations from the digitised terminus to the digitised fjord wall. Just browsing some of the terminus polygons, I noticed some that still had what I judged to be quite long extrapolations. For example, Jakobshavn Isbrae/Sermeq Kujalleq on 2002-05-30 has a ~4.5 km completely straight line and 2004-10-03 has a ~6 km straight line, which I assume are extrapolations. I assume this is occurring because the fjord widens a lot as the terminus has retreated, so even short digitisation along wide sections of fjord, could be as long or longer than the minimum fjord width further seaward. One alternative approach that comes to mind is to compare the length of the digitisation to the distance to the nearest fjord wall – if the remaining distance is only a small fraction of the length, then extrapolation may be ok. Actually, if one end of the digitisation intersects the fjord wall, then you could even compare the straightline distances between the unfinished end point and both fjord walls to determine whether extrapolation is justifiable. I deliberately chose this glacier because I thought it would be tricky, as the width of the fjord changes a lot and the glacier has bifurcated, so these might be isolated issues.

4. Linking broken digitisations: I can see that you remove multi-part delineations in stage 1 of your filtering, but I found some examples (e.g. Jakobshavn Isbrae/Sermeq Kujalleq on 2005-03-03) where it looked like linking digitisation segments was done in the wrong order, resulting in the northern part of the digitisation being extrapolated to the southern fjord wall, creating lots of self-intersections and probably a very large apparent terminus advance. It may be worth checking that stage 1 is working, or perhaps throw in some checks for large gaps between segments would help to get the order correct. Or at least check for large gaps after linking everything would help you to discard this type of failure.

5. There are some terminus digitisations that appear to go outside of the fjord walls (e.g. Jakobshavn Isbrae on 2015-03-04). I assume this is because the walls were digitised before all of the terminus traces are available. Is this likely to be consequential for the measurement of frontal ablation? If not, then I think it would be acceptable to retain or exclude these as you see fit, rather than modify the algorithm or redraw the fjord walls.

Removal of delineations with fjord walls: I was surprised to see that you remove the PROMICE delineation in Figure 3. I couldn't find the corresponding part of the text that described that step, but it looks like an important choice. I don't know how many delineations that step removes, so perhaps it makes little difference, but wouldn't it be straightforward to include delineations of that type and just clip them when they intersect the fjord wall (which you are doing anyway)?

Use of velocity to remove traces that have advanced too far: I really approve of this step, but I noticed some digitisations that had advanced unrealistically far, so I wonder if further cleaning is required. For example, at Jakobshavn Isbrae, there is a ~12 km advance along the southern fjord wall from 2007-03-08 to 2007-04-05, which seems unrealistic to me. The centreline does not advance so much, which is presumably why the algorithm retains the latter trace. I wonder if setting a maximum rate of change in area for each glacier, based on the long-term average, width-averaged near-terminus velocity would help to identify and remove these traces?

- A minor point on this step: I found the use of "midpoint" and "centreline" a bit confusing. Presumably, the intersection of a central flowline with a trace is not always the midpoint of the trace?

Line 186/7: As written it sounds like a trace on, say, the 30[th] of the previous month would not be used even if the next month did not have a trace until, say, the 10[th]. Is that right? I'm also not sure I follow this logic, because it implies that delineations closest to the 1[st] of each month would somehow be more accurate than other traces. Why not just say that the accuracy of the measurements is too low to resolve short-term (<1 month) changes in terminus position? If that is the intended meaning here.

Line 191/2: Can you clarify what is meant by "subsequent temporal averaging"? It sounds like averaging of terminus positions, but I'm not sure that is intended.

Line 235: "quality control and temporal filtering". Do these refer to the steps described above? If so, I can't see where temporal filtering comes in.

Line 247: "nearest point on the polygon boundary". Just looking at one polygon boundary (Jakobshavn), the points are spaced quite far apart. Since you connect the terminus delineation with polygon points, not polygon edges, this means the terminus doesn't always go to the closest part of the fjord. I wonder if it would be worth making your polygon point spacing smaller, to improve that connection. I have found combining arclength and interparc quite helpful for this kind of thing:

Line 249: Somewhere further up I mention some traces that have quite long extrapolations to the fjord edges and suggest a comparison like this to reduce those. Since you already have it implemented, it may be worth trying with some different thresholds to reduce the extrapolation distance.

Methods – surface elevation change/ice thickness

Line 127 (data sources): "glacier specific surface elevation change rates" – without checking the dataset from Khan (2017), it's not clear what this means. Are they scalar values for each of the study glaciers representing the average rate of change over that period, or are they time-series of elevation change rates representing rates over certain (ir)regular epochs? Are they observations of elevation change, or based on fits through the data? Are they averaged across the whole basin, or some smaller area near the terminus? If so, did you do that here or is that how the data come from Khan (2017)? If you do retain this section describing the data sources, then I think it would be worth clarifying these points.

Line 128 (data sources): "latest ArcticDEM image...": Perhaps some of this becomes clear in the methods, but I'm having trouble working out how it can be a single 'latest' image, that also covers the most advanced and retreated position, and that also covered the full extent of the tidewater glacier. Presumably you needed at least two images per glacier and picked images acquired closest-in-time to when each glacier was at its most advanced or retreated? The same goes the AeroDEM. I think you should clarify which ArcticDEM dataset was used here – the strips or the mosaics, and at what resolution? I think ArcticDEM begins in 2007, so if I understand correctly, you have an 18-year period (1988-2006) with no surface elevation observations – at least, that is how it appears from the short statement in the data sources section, which is one of the reasons I suggest rethinking that section.

Line 255: Careful with "ice surface elevation" vs "ice surface elevation change". As written, it's not clear which you have from Khan (2017).

Use of ArcticDEM and AeroDEM: I have some concerns about the direct use of these datasets. The ArcticDEM is an excellent product, but I do not think that the strips are georeferenced sufficiently well nor erroneous height measurements sufficiently well removed to use the strips as-is in a straight DEM differencing. I am not familiar with the AeroDEM, but these may suffer from similar imperfections. In this case, the thickness changes may be a small fraction of the volume and mass change of the glacier, given the small integration area, but that should be quantified – if you applied no thickness changes, how would that affect your measurements? Nevertheless, I don't think it is justifiable use the raw strips and you should either perform some corrections on the strips to remove erroneous data, remove vertical biases and ensure proper alignment, or you should use some alternative elevation or elevation change dataset.

- A related but straightforward question: Are the AeroDEM and ArcticDEM relative to the same geoid?

Line 269: "the SCR is" – can you clarify where the SCR comes from in this case if no Khan (2017) data are available.

Line 276: can I check that you ensured that the ice surface and bed topography data were relative to the same geoid?

Line 279: this point is only really relevant to Jakobshavn Isbrae up to 2003 - when calculating the volume of floating ice, how did you determine the ice thickness?

In this section, you draw on a range of datasets with very different spatial resolutions. Can you clarify which datasets were up or downsampled and to what resolution, in order to perform these steps?

Methods – uncertainty quantification

Line 316: statement re ice density uncertainty is repeated on line 314

Line 326: a 30 meter error for the delineated part seems acceptable, but some of the polygons contain long segments that were extrapolated, not delineated. I think that it would be worth retaining information about the length of the extrapolations and incorporating that into the error estimate somehow. Alternatively, you could erode your polygons by 1 or more pixels and calculate the resulting change in area.

Equation 9 and preceding text: shouldn't this just be the root-sum-square of the mass uncertainties at $t_1$ and $t_2$? Where the mass uncertainty is the root-sum-square of the uncertainties in the length, width and thickness estimates at each time.

- As written, the thickness uncertainties are not described (at least not here). They should account for the uncertainties in the reference surface elevation, surface elevation change between the reference time and measurement time, and bed elevation.

Results

Paragraph starting line 360: it would be nice to see a comparison of the long-term average discharge and long-term average frontal ablation for all of the glaciers combined and individually, either/or as numbers of a graphic. There are some glaciers listed here that don't appear in the top contributors list of Mankoff et al. (2020), so the inclusion of mass change owing to terminus advance/retreat is clearly affecting the picture of solid freshwater export from the ice sheet, but it would be great to put numbers on that. This would really support the statement on line 374 that include terminus changes increases frontal ablation – I'm sure it does, but you should use your dataset to demonstrate it.

Line 373: does this imply that including terminus position change does not affect frontal ablation on longer time-scales? As with the above point, including the numbers somewhere in the main text will demonstrate it either way.

Line 397: "values which have to be summed…" – since you have a time-series of terminus area, you could directly calculate the difference between some appropriate terminus at the start and end of each decade, then you wouldn't have to accumulate the errors through time. I appreciate that is not the 'product' being published here, but it would be a fairer comparison. Writing this made me realise that the comparison with Kochtitzky et al. (2023) is also quite difficult because termini can move quite quickly over short times, so there is no such thing as a decadal average terminus position change – it will always be a comparison of snapshots at two times, regardless of how far apart they are. With that in mind, it might just be worth comparison your termini with theirs, as that is likely the largest source of error.

Line 408: I can't quite work out what you mean by "spatial variability between observations" in this context. Is this to aid comparison between glaciers at some point? Can you rephrase this?

Line 404: "any tidewater glacier" in Greenland or anywhere globally?

In this list starting on 405: don't you also need surface elevation change data and discharge?

Conclusion

Line 420: it is good to see a statistic here, but from the data presented in the manuscript, it is hard to evaluate how representative or whether it is an extreme case that occurred at one time on one glacier. It would be great if you could quantify how much the inclusion of terminus position change affects frontal ablation for each glacier and all glaciers combined over seasonal, annual and longer time-scales.

**Technical corrections**

Line 165: comma after "Here"

Line 184 and line 194: geolocation of georeferencing?

Line 188: 1986-present. Check correct because the data and abstract show 1988-2018/2020?

Line 381: odd use of brackets in the citation

Line 383: remove "however" and replace comma with semi colon (I think…)

Line 442: check the unit on "terminus mass change" in the glossary and elsewhere

Figure 4 caption: "$m_2$ is to the right of the normal n" – it is when viewed from that perspective, but can you be certain that is always how it is being implemented in the code when you have glaciers flowing in different directions? Would the central flowline be a useful reference point from which to determine direction here?

Figure 8 and equivalent panels in the SI: I find the lines in this panel very hard to read at 100%. Can you make them thicker and produce the plot at the intended display size, if you aren't already? The max error text is also impossible to read at 100 %.

Figure 9: check for font size consistency

Figure 9: are the circles representing 14-20 % identical? I cannot tell the difference.

Figure 10: Perhaps it's personal preference, but I think this plot would be clearer if you (1) included the difference between the estimates and (2) sorted the glaciers according to that difference (or sort by frontal ablation magnitude).

Figure 10: I find the bars quite hard to see – could you make them wider by using up some of the white space between glaciers?

Figure S4: This may be more than a technical correction. I was surprised to see a glacier included that did not include discharge, given the contribution of discharge to frontal ablation. I would recommend removing this glacier and others like it, or perhaps including them separately somehow.

Benjamin Davison

---

## Referee Comment (RC2)

**Review comments for *A Frontal Ablation Dataset for 49 Tidewater Glaciers in Greenland***

**General comments**

To date, most frontal ablation estimates of the Greenland Ice Sheet have not considered mass change due to calving front variations. The studies that have, however, are either limited in spatial coverage or temporal resolution. Therefore, the new frontal ablation dataset produced in this study represents a significant step forward in achieving the Greenland-wide high-resolution frontal ablation assessment and improving the mass balance quantification, especially the processing framework presented in this study has the potential to be directly applied to other tidewater glaciers. This study is timely and useful and will be beneficial to different cryosphere communities including remote sensing and ice sheet modelling. However, several major issues need to be addressed before I can recommend it for publication in ESSD.

**Specific comments**

1. **Dataset**:

    a. The currently published dataset contains too many separate folders - one geopackage for one data type of one glacier, this makes it difficult to visualize/use this dataset. The advantage of geopackage over shapefiles is that it can have multiple layers with each layer having a different data type. Please consider merging different layers of different glaciers in one geopackage.

    b. I see that frontal ablation plots for all the studied glaciers are presented in the supplementary file, will it be possible for the authors to provide these figures in the data product as well?

2. **Product Description**: this section is not a description of the data product generated in this study, it is purely about the frontal ablation calculation methods, therefore it should be better placed in the Methods section. Here please give an overview of all the relevant information that we need to know about the dataset itself. What kind of data product you have produced? What is the spatial coverage and temporal resolution of this data product? What data files are included in the data product?

3. **Data Sources and Methods:**

    a. In the Data Sources section, many places mention "see Methods", this is not helpful. Please label each subsection in Methods, so you can easily cross-reference different data sources and the corresponding processing steps.

    b. Please move Table S1 to the main text, this table is important for readers to know about the spatiotemporal resolution and quality of each input data source. Also in this table, note that temporal resolution is different from temporal coverage and please clarify accordingly. Please include the Khan (2017) data, which is an important data source for adjusting ice thickness according to the manuscript. The data type of ArcticDEM used in the study, and the time periods of both ArcticDEM and AeroDEM, should also be provided in this table.

    c. Methodology needs to be restructured. First, please provide a flowchart to give an overview of all the major processing steps outlined in this section. This can help readers understand the general processing workflow. Second, there are many nice sketch figures presented in the method section, while they are helpful for readers to visualize each processing step, most of them can be easily merged into one single integrated figure, such as Figures 2/3/5/7. For Figure 5 and Figures 7B-E, I

am not even sure whether they are necessary in the main text as these details are trivial concepts that are easy to understand from the text description alone, perhaps it's better to put them in the supplementary material.

4. **Surface elevation change and ice thickness:**

   a. Khan (2017) dataset: in line 128 it says the time period of Khan dataset is 1995-2015. However, from the data link you provided in the manuscript (https://dataverse.geus.dk/dataset.xhtml?persistentId=doi:10.22008/FK2/GQJJEA), the Khan data covers the period from 2011 to 2020, which is correct? The more urgent issue is why not use the latest Khan (2023) dhdt dataset? Won't this latest dhdt improve the ice thickness calculation in this study? Again, there is no introduction of Khan dhdt dataset in the Data Sources section. Why did you choose to use Khan dhdt instead of other dhdt products? What is the advantage of this data product and how it was generated? Please also cite the associated publication for the Khan dataset.

   b. In Figure 6, if Khan dhdt is already available for the date of the terminus change, doesn't this mean that TOD has already been included in the K-SCR time range? Why there is still an additional step of checking whether TOD is in K-SCR time range?

   c. ArcticDEM and AeroDEM: what type of ArcticDEM is used in the methods? Are they mosaics or strips? If using the strip and it covers the selected terminus trace, is it still necessary to adjust elevation based on dhdt?

5. **Results section:**

   a. Figure 9 gives an overview of the frontal ablation rates produced in this study. The choices of different number scales of the annual mean frontal ablation look a bit random, why not use integers here or a colorbar that can clearly show the value variations? Can you also compare the frontal ablation rates with the Mankoff ice discharge data statistically, ideally in a histogram? This can show the impact and necessity of including terminus changes in the frontal ablation calculation.

   b. Figure 10 and Line 380-383. Authors claimed that "agreement is reduced for the period 2010-2020", this is not obvious from Figure 10, although Table S3 provides a comparison. I recommend calculating the difference in frontal ablation for each glacier between this study and Kochtitzky et al. (2023) dataset, then plotting these differences in a map similar to Figure 9, or in a histogram.

   c. Line 398-400, it briefly touched on the issue of ice discharge but only gave one example. Can you compare the discharge data used in Kochtitzky et al. (2023) and the Mankoff ice discharge data for all the studied glaciers? This should help clarify if the discrepancy in frontal ablation rates is from terminus change or ice discharge.

   d. Line 346: This sharp increase in frontal ablation (Figure 8A) started in 2005/2006, not 2004/2005. There is significant interannual variability in frontal ablation and given there is no errorbar provided in the plot, it is difficult to claim that this increase in frontal ablation is from changes in velocity and terminus, especially since there is also a sharp change in 2010/2011 according to Figure 8A.

   e. Line 348-351: can you fit linear regression before and after 2004/2005 to see if there is actually an increase in frontal ablation before and after the velocity/terminus changes?

> f. In Figure 8 and Figures S2-S48, the temporal intervals in the x-axis are not regular – it contains both two-year and three-year intervals but they have the same width, is this a plotting error?

**Technical comments**

1. Abstract needs rewrite. For example, BedMachine has been mentioned twice and please be clear in Line 34 this is consecutive *calving front* observations.

2. Line 38-39: "any tidewater glacier". Is it only for the Greenland Ice Sheet? Or globally?

3. Line 69-73: this paragraph reads like a separate statement and feels very sudden here. Since this is a dataset paper for a dataset journal, the focus be the data product itself instead of the methodology, please recheck the journal submission guideline.

4. Line 105-106: could you please provide a figure on the spatial coverage of the Bedmachine v4 bathymetry data sources in the supplementary file?

5. Line 127: as mentioned in the major comments, please provide detailed information about the Khan dhdt dataset here.

6. Line 142: "5 km upstream of the terminus" is this the most retreated terminus location?

7. Line 173: please be clear that the individual terminus positions are compared to each other to get area changes.

8. Line 182 **Terminus Positions**: I suggest introducing terminus positions first before talking about fjord boundary, because in fjord boundary section there are lots of descriptions on how to change the terminus direction without knowing what terminus data were used.

9. Line 188-191: this sentence is difficult to understand, please rephrase. Can you give a number on the maximum level of uncertainty involved?

10. Line 191: what is the threshold for time difference?

11. Line 212: where is this centerline from?

12. Line 217: change "mean value velocity" to mean velocity.

13. Line 229-232: please move this information to Line 103-110 when first mentioning how these 49 glaciers were selected.

14. Line 269: please be clear about "the SCR", is this "AA-SCR"?

15. Line 357-359: please be more explicit about the importance of having this higher variability in frontal ablation. Or why does it matter to have this variability compared to ice discharge that doesn't have it?

16. Figure 2: please label the upper and lower fjord walls in the figure.

17. Figure 3 and Line 204-205: Can you explain a bit about how these delineations can skew the mass changes? It is not clear from Figure 3 at all.

18. Figure 8: In Line 351-355, the figure labels are all wrong here, and there is no Figure 8E, please carefully check the figure labels throughout the manuscript. It is hard to distinguish

between TMC and F in Figure 8A given the current choices of linewidth and line color. In Figure 8C, which direction is terminus retreat or advance?

---

## Author Comment (AC1)

We thank Benjamin his thorough and helpful review of the manuscript. His comments have provided valuable input to improve the manuscript, the processing chain and the data product. We have made significant changes to the manuscript, processing chain and data product, with detailed responses to the reviewer comments below. Reviewer comments are shown in black, our responses in blue. All line numbers in our responses refer to the revised manuscript.

**General comments**

This manuscript describes the method used to generate a new dataset of frontal ablation for a sample of Greenland's tidewater glaciers. The dataset and workflow will be a great asset to the community and I am confident that the products will enable lots of good science. Overall, the method appears sound but I think the pre-processing of the terminus delineations and the estimate of ice thickness changes need to be improved to avoid erroneous changes in glacier volume. The majority of the manuscript is used to describe the method for calculating mass changes owing to terminus advance and retreat – I think more attention needs to be devoted to describing and documenting the dataset itself, and to quantifying the impact of various choices through the processing chain. In particular, the contribution of advance/retreat to frontal ablation should be quantified for all glaciers, individually and combined, at a range of temporal scales. I think the error metric for terminus mass change needs to be improved, particularly through the inclusion of thickness errors.

> We'd like to thank the reviewer for the time and effort to conduct such a thoughtful and detailed review. We have aimed to address the general comments above, with details provided in our responses to the specific comments below.

**Specific comments**
**Data product:**
The errors in D and TMC should also be provided, so that users can see how they contribute to the error in F.

- > Good point. We have now included the error in D and TMC into the final data product.

I think it would be helpful if the time-series for each glacier were combined into a single file, perhaps a NetCDF. This would make it much easier for potential users to amalgamate the measurements from different glaciers.

- > The dataset is now available as single shapefile, geopackage and NetCDF.

A few issues that I assume are minor:
For Akullersuup Sermia, the last two entries for TMC and F are empty, but D and Max_Error_F are provided. Is that expected?

- > We checked the dataset, and it appears that this was an error that occurred during file conversion.
- > We ensured that data for all glaciers is consistent.

Time period: Again for Akullersuup Sermia, data are available for the period 1988 to 2018. In the manuscript, the abstract states 1987-2020, the caption of Figure 9 states 1987-2015 but

line 361 of the submitted version states 1987-2018. Do different glaciers have different temporal coverage, with some (a few or most?) extending to 2020? If the glaciers do have different time periods, it would be easier for users if you at least make all of the time-series the same length by padding with NaNs as necessary.

- Different glaciers have different temporal coverage, coming from the terminus position availability. The longest span we have for any glacier is 1987-2020, but for the purpose of Figure 6 (formerly Figure 9), we use the period 1988-2018 to ensure we can include all glaciers in the plot. In the figure and figure caption we have noted a few glaciers for which we don't have data for the full 1988-2018 period required by the plot.
- Following your suggestion, we have made all data consistent to cover the period 1987-2020 in the final dataset by padding with NaNs.

**Data sources:**
Initially, I found this section a little confusing. It reads somewhat like an abstract for the methods, giving a very brief overview but without much detail, some of which is repeated again in the methods. The purpose of this section is just to tell the reader what data you used – giving them some identifier or name – so that you can refer to them unambiguously in your methods. With that in mind, I wonder if this section might be better structured as a series of bullet points where you clearly state the parameter (terminus position, elevation, ice velocity, discharge) and what data source(s) were used, then bring Supplementary Table into the main text at this point. I would recommend adding the time-period to the table (and possibly to the text describing the data).

- Thank you for this suggestion - we have changed this section of the manuscript to bullet points as suggested by the reviewer.
- We added the suggested further details to the former Table S1 and included this table in the main text (now Table 1).

**Methods – fjord geometry and terminus positions**
Line 169: lower/upper fjord wall is hard to understand. Should it be southern/northern? If so, then is the code able to deal with glaciers flowing north to south or south to north? Could you use true left or right instead – that would be independent of glacier orientation, but it would entail bringing in the flow direction data earlier, so might not be worth the additional effort.

- The code is able to deal with glaciers flowing in any direction. We changed "upper" and "lower" fjord wall to fjord boundary 1 and fjord boundary 2. We hope that this clarifies that these are independent of glacier flow direction.

**Fjord wall delineation and polygon creation:** overall, this approach for creating polygons seems very robust, but I have a few queries and have spotted a few areas where the algorithm appears to struggle:

1. For ice streams like Jakobshavn Isbrae/Sermeq Kujalleq, where there is no clear lateral topographic boundary in the vicinity of the present-day terminus, how did you determine where to place the fjord walls?

- This is a good question – we did this on a case-by-case basis – mostly you can for example see strong shear margins at the edge of fast flowing ice and these provide good boundaries.

2. As glaciers thin, they can reduce in width. Depending on the date of the image used to digitise your fjord walls, you might be slightly (and it probably is only very slightly) overestimating or underestimating the change in area owing to terminus position change in

years far before or after the image was acquired. I think it would probably be more effort than it's worth to quantify the impact of this because the associated error is likely smaller than other errors, but I suggest that this effect is acknowledged somewhere.

- Agreed - we do not take lateral thinning of the glacier into account, but we feel the error introduced is minimal because presumably these areas have small ice thicknesses. We added a sentence acknowledging this possibility to the manuscript in section 2a [Lines 196-198].

3. Definition of terminus width when filtering polygons: As I understand it, the purpose of this step is avoid making long extrapolations from the digitised terminus to the digitised fjord wall. Just browsing some of the terminus polygons, I noticed some that still had what I judged to be quite long extrapolations. For example, Jakobshavn Isbrae/Sermeq Kujalleq on 2002-05-30 has a ~4.5 km completely straight line and 2004-10-03 has a ~6 km straight line, which I assume are extrapolations. I assume this is occurring because the fjord widens a lot as the terminus has retreated, so even short digitisation along wide sections of fjord, could be as long or longer than the minimum fjord width further seaward. One alternative approach that comes to mind is to compare the length of the digitisation to the distance to the nearest fjord wall – if the remaining distance is only a small fraction of the length, then extrapolation may be ok. Actually, if one end of the digitisation intersects the fjord wall, then you could even compare the straightline distances between the unfinished end point and both fjord walls to determine whether extrapolation is justifiable. I deliberately chose this glacier because I thought it would be tricky, as the width of the fjord changes a lot and the glacier has bifurcated, so these might be isolated issues.

- We agree that there were issues with the delineations at Jakobshavn Isbræ.
- After further investigation of this glacier, we found that one operator created delineations that do not cover the whole terminus, but rather traced the northern and southern branch of Jakobshavn Isbræ separately, which explains the long extrapolations in this case.
- We manually removed these particular delineations, and after doing that, feel that the remaining process is sufficiently robust to exclude long extrapolations.
- The manual removal of these delineations is now noted in manuscript in section 2b [Lines 200-203]:
  *"The TermPicks delineations of all investigated tidewater glaciers are first visually examined to identify obvious outliers caused by e.g., false georeferencing of the satellite image, delineation of mélange or, in the case of Jakobshavn Isbræ, delineations of only one branch of the terminus."*

4. Linking broken digitisations: I can see that you remove multi-part delineations in stage 1 of your filtering, but I found some examples (e.g. Jakobshavn Isbrae/Sermeq Kujalleq on 2005-03-03) where it looked like linking digitisation segments was done in the wrong order, resulting in the northern part of the digitisation being extrapolated to the southern fjord wall, creating lots of self-intersections and probably a very large apparent terminus advance. It may be worth checking that stage 1 is working, or perhaps throw in some checks for large gaps between segments would help to get the order correct. Or at least check for large gaps after linking everything would help you to discard this type of failure.

- Thank you for noting this. We believe these issues have been resolved by excluding the separate delineations as described in the previous comment.

5. There are some terminus digitisations that appear to go outside of the fjord walls (e.g. Jakobshavn Isbrae on 2015-03-04). I assume this is because the walls were digitised before all of the terminus traces are available. Is this likely to be consequential for the measurement

of frontal ablation? If not, then I think it would be acceptable to retain or exclude these as you see fit, rather than modify the algorithm or redraw the fjord walls.

- We are not quite sure what the reviewer is referring to here – certainly some of the raw terminus traces go beyond the fjord walls, but after our processing the end of the terminus trace should coincide with the fjord wall. We double checked the delineations after they have been clipped/extrapolated to the fjord walls and couldn't find any that go beyond the delineated fjord walls. It's possible this is related to the two branches problem with Jakobshavn Isbræ described in the last two comments that has now been fixed.

**Removal of delineations with fjord walls**: I was surprised to see that you remove the PROMICE delineation in Figure 3. I couldn't find the corresponding part of the text that described that step, but it looks like an important choice. I don't know how many delineations that step removes, so perhaps it makes little difference, but wouldn't it be straightforward to include delineations of that type and just clip them when they intersect the fjord wall (which you are doing anyway)?

- The PROMICE delineation shown in Figure 3 is excluded because the trace consists of 4 separate line segments, and this is stated in section 2b [Lines 222-226] The problem is that it is programmatically not trivial to determine which of the four lines is the actual terminus we're interested in without manually checking.
- Note that this does not mean that we exclude all PROMICE delineations, but rather those which contain more than one line.

Use of velocity to remove traces that have advanced too far: I really approve of this step, but I noticed some digitisations that had advanced unrealistically far, so I wonder if further cleaning is required. For example, at Jakobshavn Isbrae, there is a ~12 km advance along the southern fjord wall from 2007-03-08 to 2007-04-05, which seems unrealistic to me. The centreline does not advance so much, which is presumably why the algorithm retains the latter trace. I wonder if setting a maximum rate of change in area for each glacier, based on the long-term average, width-averaged near-terminus velocity would help to identify and remove these traces?

- Unfortunately, the two-branch issue at Jakobshavn Isbrae does seem to have caused multiple issues and this is one of them – thank you for spotting this. This issue should now be resolved with the exclusion of delineations that cover different branches of the terminus.
- Beyond this, we feel that the existing method is sufficient, but we hope to implement your suggestion in a future iteration because it could require less data download on behalf of the person running the code.

A minor point on this step: I found the use of "midpoint" and "centreline" a bit confusing. Presumably, the intersection of a central flowline with a trace is not always the midpoint of the trace?

- We renamed the points to intersection points ($c_1$ and $c_2$) and clarified how they are derived in the text [Lines 249-259].

Line 186/7: As written it sounds like a trace on, say, the 30th of the previous month would not be used even if the next month did not have a trace until, say, the 10th. Is that right? I'm also not sure I follow this logic, because it implies that delineations closest to the 1st of each month would somehow be more accurate than other traces. Why not just say that the accuracy of the measurements is too low to resolve short-term (<1 month) changes in terminus position? If that is the intended meaning here.

- Thanks for asking for clarification here. You're correct that the key point is that estimates of frontal ablation at higher temporal resolution than approximately monthly are unreliable because the error on the ablation increases as the time interval decreases and because the error on the terminus delineations becomes more prominent as the time interval decreases (see e.g. Methods section 2f). Therefore, we wanted to pick delineations that were spaced at an approximate interval (which in the code can be set to monthly, 3-monthly or annual; the results in the paper show 3-monthly). We chose to pick delineations that were close to the first day of the time period (because that is as good a choice as any), but you're correct that we further specify that they are after the first day. In hindsight it may have been better to simply choose the closest to the first (so that something from the 30$^{th}$ of the previous month would be prioritised over the 10$^{th}$ of the next month), but in practice the delineations are sufficiently abundant that this doesn't make much difference, and also where they are sparse we still are successful at finding terminus traces spaced at approximately our requested interval. We have clarified some of this motivation in the text (Methods section 2b).

Line 191/2: Can you clarify what is meant by "subsequent temporal averaging"? It sounds like averaging of terminus positions, but I'm not sure that is intended.

- Apologies that this was unclear in the text – perhaps our writing was confusing on the point of whether we are choosing a monthly or three-monthly time period. We have revised this paragraph to be clearer and removed the phrase in question.

Line 235: "quality control and temporal filtering". Do these refer to the steps described above? If so, I can't see where temporal filtering comes in.

- We rephrased the sentence to clarify what we mean and connect it to the processes described in the previous steps. It now reads [Lines 267-269]:
  *"Overall, after quality control and sub-setting the dataset to one observation per month, the dataset contains 36.8 % of all terminus delineations (6698 of 18202) for the selected 49 glaciers."*

Line 247: "nearest point on the polygon boundary". Just looking at one polygon boundary (Jakobshavn), the points are spaced quite far apart. Since you connect the terminus delineation with polygon points, not polygon edges, this means the terminus doesn't always go to the closest part of the fjord. I wonder if it would be worth making your polygon point spacing smaller, to improve that connection. I have found combining arclength and interparc quite helpful for this kind of thing:

- The manually delineated fjord walls contain ~1400 individual points on average, and in the case of Jakobshavn Isbræ, the fjord wall coordinates are spaced approximately 30 metres apart. We feel this is sufficiently dense that the terminus trace should be extrapolated to the closest part of the fjord wall, so we're not quite sure what the reviewer is referring to here. Possibly again this is related to the issue with Jakobshavn Isbræ having two branches.

Line 249: Somewhere further up I mention some traces that have quite long extrapolations to the fjord edges and suggest a comparison like this to reduce those. Since you already have it implemented, it may be worth trying with some different thresholds to reduce the extrapolation distance.

- Thank you for this suggestion - while we agree that different methods or thresholds could be implemented, we are convinced from our various iterations of the processing chain that this would only marginally change the results.

**Methods – surface elevation change/ice thickness**
Line 127 (data sources): "glacier specific surface elevation change rates" – without checking the dataset from Khan (2017), it's not clear what this means. Are they scalar values for each of the study glaciers representing the average rate of change over that period, or are they time-series of elevation change rates representing rates over certain (ir)regular epochs? Are they observations of elevation change, or based on fits through the data? Are they averaged across the whole basin, or some smaller area near the terminus? If so, did you do that here or is that how the data come from Khan (2017)? If you do retain this section describing the data sources, then I think it would be worth clarifying these points.

- The Khan (2023) dataset (previously the Khan (2017) dataset) is an annual, 1 x 1 km gridded product of surface elevation change rate derived from Cryosat-2, IceSat-2 and Nasa ATM flights. The time resolution of the data that is input to the dataset is higher than annual, and a trend line is fit to the data within each 1 x 1 km square and the annual values extracted from this. This processing is done by Khan and not done in our paper. The processing that is done in this paper is to extract the mean of the Khan values that lie within the polygons we have constructed. We've edited the relevant paragraph (now section 2d) to make this clear.

Line 128 (data sources): "latest ArcticDEM image...": Perhaps some of this becomes clear in the methods, but I'm having trouble working out how it can be a single 'latest' image, that also covers the most advanced and retreated position, and that also covered the full extent of the tidewater glacier. Presumably you needed at least two images per glacier and picked images acquired closest-in-time to when each glacier was at its most advanced or retreated? The same goes the AeroDEM. I think you should clarify which ArcticDEM dataset was used here – the strips or the mosaics, and at what resolution? I think ArcticDEM begins in 2007, so if I understand correctly, you have an 18-year period (1988-2006) with no surface elevation observations – at least, that is how it appears from the short statement in the data sources section, which is one of the reasons I suggest rethinking that section.

- We have three datasets here: AeroDEM (dates vary depending on region but in range 1978-1987), surface elevation change rates from 1995-2020 (Khan, 2017) and 2m ArcticDEM strips (dates from 2007 but we only use the latest available strip). Therefore, for every glacier in our dataset, we have at least a single DEM in the period 1978-1987 (AeroDEM) and a single DEM from around the past 5 years (the latest available ArcticDEM strip). We can use these 2 DEMs to linearly interpolate for the surface elevation at any time between the DEM acquisition dates (this obviously assumes constant surface elevation change rate). If we require the surface elevation outside of the time between the 2 DEMs, we extrapolate using the same constant surface elevation change rate. If we additionally have surface elevation change rates from Khan (which we don't have for every glacier due to incomplete spatial coverage in their data) then we can replace some of the constant surface elevation change rate in the linear interpolation with the Khan data. We hope this is now clear. You are correct that for some glaciers (those without Khan data), we use no surface elevation observations between the date of the AeroDEM and the latest ArcticDEM strip. We could have used multiple ArcticDEM strips to improve the temporal resolution here, but the temporal variability in the ice thickness has only a small impact on the frontal ablation. Variability in frontal ablation is really driven by the length of terminus advance or retreat, rather than the much smaller and slower change in the ice thickness of the terminus region, and furthermore uncertainty in the bedrock elevation dominates over the possible change in ice thickness (see next response for

a specific example of this). We have revamped the section in question (now section 2d) and hope this is all now clearer.

Line 255: Careful with "ice surface elevation" vs "ice surface elevation change". As written, it's not clear which you have from Khan (2017).

- We changed the wording to clearly differentiate between surface elevation and surface elevation change.

Use of ArcticDEM and AeroDEM: I have some concerns about the direct use of these datasets. The ArcticDEM is an excellent product, but I do not think that the strips are georeferenced sufficiently well nor erroneous height measurements sufficiently well removed to use the strips as-is in a straight DEM differencing. I am not familiar with the AeroDEM, but these may suffer from similar imperfections. In this case, the thickness changes may be a small fraction of the volume and mass change of the glacier, given the small integration area, but that should be quantified – if you applied no thickness changes, how would that affect your measurements? Nevertheless, I don't think it is justifiable use the raw strips and you should either perform some corrections on the strips to remove erroneous data, remove vertical biases and ensure proper alignment, or you should use some alternative elevation or elevation change dataset.

- These are excellent points – thank you for the attention to detail. We've got two related responses here. Firstly, as a quick check we compared ArcticDEM and AeroDEM by randomly sampling bedrock area (5-20 points per glacier); across all glaciers in the dataset, the mean difference is 4.4±20.6 metres, suggesting there are not huge biases or offsets in the DEMs. Secondly, our feeling is that the huge effort (data volume, time, expertise) of performing corrections on the DEM strips would simply not be justified due to the very likely limited impact on the results. To give a concrete example, consider that Helheim has a rough ice thickness of 700 m, and a bedrock elevation uncertainty of approximately 70 m (Table S1). Even if a DEM is out by 20 m systematically, this is much smaller than the bedrock elevation uncertainty and only approximately 3% of the ice thickness. Plus, the frontal ablation varies by roughly +/-100%, the vast majority of which can only be due to terminus position change (Figure 5). As such, we do appreciate the suggestion but respectfully suggest it is not worth the investment in this particular case.

A related but straightforward question: Are the AeroDEM and ArcticDEM relative to the same geoid?

- Yes, all datasets are referenced to the same ellipsoid.

Line 269: "the SCR is" – can you clarify where the SCR comes from in this case if no Khan (2017) data are available.

- This section has now been rewritten (see response to comment above).

Line 276: can I check that you ensured that the ice surface and bed topography data were relative to the same geoid?

- Yes, again, all datasets are referenced to the same ellipsoid (see above comment).

Line 279: this point is only really relevant to Jakobshavn Isbrae up to 2003 - when calculating the volume of floating ice, how did you determine the ice thickness?

- In fact, we did not account for the fact that the front of Jakobshavn Isbræ was floating before 2003. Thank you for pointing this out. Since the concept of frontal ablation is rather different for a floating ice tongue compared to a grounded glacier, we have decided it is best to remove our estimates for Jakobshavn Isbræ before 2003. This is stated in the paper in the product description section [Lines 103-104].

In this section, you draw on a range of datasets with very different spatial resolutions. Can you clarify which datasets were up or downsampled and to what resolution, in order to perform these steps?

- It's now stated in section 2d that we resampled the datasets to the BedMachine v4 grid (150 m resolution). Depending on the initial resolution of the data (given in Table 1), this is either an up-sampling or a down-sampling.

**Methods – uncertainty quantification**
Line 316: statement re ice density uncertainty is repeated on line 314

- We removed the statement in line 316.

Line 326: a 30 meter error for the delineated part seems acceptable, but some of the polygons contain long segments that were extrapolated, not delineated. I think that it would be worth retaining information about the length of the extrapolations and incorporating that into the error estimate somehow. Alternatively, you could erode your polygons by 1 or more pixels and calculate the resulting change in area.

- The extrapolated terminus traces tend to be near the lateral boundaries of glaciers where the ice is thinner, so that the impact on the frontal ablation is likely to be limited. Perhaps more convincingly, note that while the total error is time dependent, for simplicity we use the maximum error at any point in time (see end of section 2f), plus we use the centerline bedrock error as representative of the full terminus region. As such we feel the error calculation is already quite generous and the additional complexity of adding an error estimation for the extrapolated terminus traces would change the errors rather little. We hope the reviewer is convinced by this rationale.

Equation 9 and preceding text: shouldn't this just be the root-sum-square of the mass uncertainties at t1 and t2? Where the mass uncertainty is the root-sum-square of the uncertainties in the length, width and thickness estimates at each time.

- This is a subtle point – apologies that we did not make this clear in the manuscript. You would be correct if the errors in width and thickness were random, but they are systematic (because we use the same digitised fjord walls at times t1 and t2 and the same bed topography at times t1 and t2). Put another way, if we assumed the thickness error was random, there would be a term $L^2 W^2 \Delta H^2$ in the errors, but it doesn't make sense for us to account for the bedrock error over the full trunk of the glacier (L) when most of that full trunk is still there at the next timestep and the terminus mass change only depends on the bit that has retreated. In light of the systematic nature of these errors, we found the most logical way to consider them to be imagining purely the block which is the difference between V1 and V2 and doing the error analysis on that block. We have however now accounted for the error on the width of that block (which is after all the same sort of delineation error as for the terminus positions). This has been made clearer in the manuscript in [Eq. 8; Lines 370-371].

As written, the thickness uncertainties are not described (at least not here). They should account for the uncertainties in the reference surface elevation, surface elevation change between the reference time and measurement time, and bed elevation.

- We added an equation (Eq.9) to explain the calculation of ice thickness uncertainties.

**Results**
Paragraph starting line 360: it would be nice to see a comparison of the long-term average discharge and long-term average frontal ablation for all of the glaciers combined and individually, either/or as numbers of a graphic. There are some glaciers listed here that don't appear in the top contributors list of Mankoff et al. (2020), so the inclusion of mass change owing to terminus advance/retreat is clearly affecting the picture of solid freshwater export from the ice sheet, but it would be great to put numbers on that. This would really support the statement on line 374 that include terminus changes increases frontal ablation – I'm sure it does, but you should use your dataset to demonstrate it.

- We included a figure to show the combined total frontal ablation estimates vs Mankoff et al (2020) discharge to highlight the point made by the reviewer (Figure 7). You can see that the frontal ablation does exceed the discharge, particularly in the early 2000s when there were big retreats of glaciers such as Helheim. For a glacier-by-glacier perspective, see Figure S3, where it is seen that the frontal ablation exceeds the discharge for almost every glacier (this has to be the case unless the glacier has advanced). The new Figure 7 and Figure S3 are now referred to in the text.

Line 373: does this imply that including terminus position change does not affect frontal ablation on longer time-scales? As with the above point, including the numbers somewhere in the main text will demonstrate it either way.

- Good question – no, the frontal ablation exceeds the ice discharge over longer time periods too. The new Figures 7 and S3 demonstrate this, see also response to previous comment.

Line 397: "values which have to be summed…" – since you have a time-series of terminus area, you could directly calculate the difference between some appropriate terminus at the start and end of each decade, then you wouldn't have to accumulate the errors through time. I appreciate that is not the 'product' being published here, but it would be a fairer comparison. Writing this made me realise that the comparison with Kochtitzky et al. (2023) is also quite difficult because termini can move quite quickly over short times, so there is no such thing as a decadal average terminus position change – it will always be a comparison of snapshots at two times, regardless of how far apart they are. With that in mind, it might just be worth comparison your termini with theirs, as that is likely the largest source of error.

- We are not sure we agree that the comparison is difficult – although termini can move quite fast, the key point about frontal ablation is that it is the sum of the ice discharge and the terminus mass change. Suppose a glacier has a mean ice discharge over 2000-2010 of 10 Gt/yr. Suppose we calculate the terminus mass change using terminus traces on 1 January 2000 and 1 January 2010 and find 0 terminus mass change (if the terminus is in the same place). But then suddenly on 2 January 2010 the glacier loses 1 Gt in a single huge calving event. If we had used the trace on 2 January 2010 we would have an additional 1 Gt of terminus mass change, but in terms of frontal ablation over the 2000-2010 period this only adds 1 Gt/10 years = 0.1 Gt/yr, which is small compared to the ice discharge. Therefore, the sudden large calving event doesn't strongly impact the frontal ablation. So we think you can have a "decadal average terminus position change" (or at least a change

rate, which is what the frontal ablation cares about), and that this is a meaningful quantity – although the terminus position can change within minutes, the impact over the decadal timescale is muted by the fact you are averaging over 10 years and adding it to the ice discharge. The only time this would go wrong is if you had a slowly flowing glacier that had a sudden huge retreat, but this doesn't tend to happen, at least at grounded glaciers.
- Secondly, although we could calculate the decadal mean frontal ablation using terminus traces separated by a decade, we would get the same answer as summing up our three-monthly differences, so we don't think the suggestion would make a difference. Hopefully the reviewer is convinced by this rationale.

Line 408: I can't quite work out what you mean by "spatial variability between observations" in this context. Is this to aid comparison between glaciers at some point? Can you rephrase this?

- Thanks - we added a second sentence to clarify [Lines 479-481]: "*Delineations from multiple operators can introduce uncertainties due to the subjective interpretation of glacier termini in satellite images.*"

Line 404: "any tidewater glacier" in Greenland or anywhere globally?
In this list starting on 405: don't you also need surface elevation change data and discharge?

- The processing chain can be adapted to be applied globally – this has been added [Lines: 473-475].
- We agree that solid ice discharge is needed to derive frontal ablation estimates using this processing chain and added it to the list.
- Surface elevation change data is slightly less critical because most of the variability in frontal ablation comes from the terminus position change rather than the thinning, but we have already included "surface elevation data" in point 3 and believe this covers it.

**Conclusion**

Line 420: it is good to see a statistic here, but from the data presented in the manuscript, it is hard to evaluate how representative or whether it is an extreme case that occurred at one time on one glacier. It would be great if you could quantify how much the inclusion of terminus position change affects frontal ablation for each glacier and all glaciers combined over seasonal, annual and longer time-scales.

- We added Figures 7 and S3 to the manuscript and refer the reviewer to our responses on those points. We have also added more detail to the text here.

**Technical corrections**
Line 165: comma after "Here"

- Changed as suggested.

Line 184 and line 194: geolocation of georeferencing?

- Changed to georeferencing.

Line 188: 1986-present. Check correct because the data and abstract show 1988-2018/2020?

- The maximum temporal coverage of the dataset is from 1987-2020, however the temporal coverage is dependent on each glacier. This should be clearer now with Fig. 6A. We also ensured that the correct temporal coverage is referred to throughout the manuscript.

Line 381: odd use of brackets in the citation

- We amended the sentence. It now reads [Lines 446-449]:
  *"When comparing glaciers that are included in both studies, and calculating decadal mean values from our dataset, we find that for both periods the majority of our estimates (>50%) are within 0.2 Gt/yr of the uncertainty boundaries of Kochtitzky et al. (2023; Fig. 6B, 8; Table S2)."*

Line 383: remove "however" and replace comma with semi colon (I think…)

- The sentence has been removed during editing.

Line 442: check the unit on "terminus mass change" in the glossary and elsewhere

- We checked and amended the manuscript where necessary to ensure that the correct units are used.

Figure 4 caption: "m2 is to the right of the normal n" – it is when viewed from that perspective, but can you be certain that is always how it is being implemented in the code when you have glaciers flowing in different directions? Would the central flowline be a useful reference point from which to determine direction here?

- We manually checked that the code works for all glaciers in the dataset no matter in which direction they flow.
- The intersection point (previously mid-point) of terminus delineations corresponds, to some extent, to the central flowline.
- It is not quite clear to the authors how choosing a different, yet similar line, as a reference would enhance the dataset or the processing chain.

Figure 8 and equivalent panels in the SI: I find the lines in this panel very hard to read at 100%. Can you make them thicker and produce the plot at the intended display size, if you aren't already? The max error text is also impossible to read at 100 %.

- We changed the line colour and thickness in Figure 8 (Now Figure 5) and supplementary plots (Figs S4 -S54) to make them easier to read.

Figure 9: check for font size consistency
Figure 9: are the circles representing 14-20 % identical? I cannot tell the difference.

- We changed Figure 9 (now Figure 6) to make the results clearer.

Figure 10: Perhaps it's personal preference, but I think this plot would be clearer if you (1) included the difference between the estimates and (2) sorted the glaciers according to that difference (or sort by frontal ablation magnitude).
Figure 10: I find the bars quite hard to see – could you make them wider by using up some of the white space between glaciers?

- We changed Figure 10 (now Figure 6) after comments from reviewer 2 to maps.
- This should show the difference of frontal ablation estimates for each decade between Kochtitzky et al. (2023) and this study more clearly.

Figure S4: This may be more than a technical correction. I was surprised to see a glacier included that did not include discharge, given the contribution of discharge to frontal ablation. I would recommend removing this glacier and others like it, or perhaps including them separately somehow.

- We removed the glacier from the dataset.

---

## Author Comment (AC2)

We thank the anonymous reviewer for their thorough and helpful review of the manuscript. Their comments have provided valuable input to improve the manuscript, the processing chain and the data product. We have made significant changes to the manuscript, processing chain and data product, with detailed responses to the reviewer comments below. Reviewer comments are shown in black, our responses in blue. All line numbers in our responses refer to the revised manuscript.

**Specific comments**
Dataset: The currently published dataset contains too many separate folders - one geopackage for one data type of one glacier, this makes it difficult to visualize/use this dataset. The advantage of geopackage over shapefiles is that it can have multiple layers with each layer having a different data type. Please consider merging different layers of different glaciers in one geopackage.

- We changed the format of the dataset so that the complete dataset is available as a single geopackage, shapefile or NetCDF.

I see that frontal ablation plots for all the studied glaciers are presented in the supplementary file, will it be possible for the authors to provide these figures in the data product as well?

- We have uploaded the supplementary figures for each glacier to the repository.

**Product Description**: this section is not a description of the data product generated in this study, it is purely about the frontal ablation calculation methods, therefore it should be better placed in the Methods section. Here please give an overview of all the relevant information that we need to know about the dataset itself. What kind of data product you have produced? What is the spatial coverage and temporal resolution of this data product? What data files are included in the data product?

- We thank the reviewer for this point and have revised the section accordingly, now providing the salient product information.
- We moved the description of the frontal ablation calculation method to the methods section under the subsection "1. Background".
- We added a paragraph to describe the available temporal coverage, available file formats of the dataset as well as the variables that the dataset contains [Lines 83-92].

**Data Sources and Methods:**

In the Data Sources section, many places mention "see Methods", this is not helpful. Please label each subsection in Methods, so you can easily cross-reference different data sources and the corresponding processing steps.

- Thank you. We have introduced labelled subheadings in the Methods section to make it easier to refer to the relevant paragraph and amended the manuscript accordingly.

Please move Table S1 to the main text, this table is important for readers to know about the spatiotemporal resolution and quality of each input data source. Also in this table, note that temporal resolution is different from temporal coverage and please clarify accordingly. Please include the Khan (2017) data, which is an important data source for adjusting ice thickness according to the manuscript. The data type of ArcticDEM used in the study, and the time periods of both ArcticDEM and AeroDEM, should also be provided in this table.

- We have moved Table S1 to the main text (now Table 1), included the Khan (2017,2023) dataset, and clarified the temporal coverage / resolution.
- We included the temporal coverage of Arctic- and AeroDEM in Table 1. The exact timestamp for each DEM can be derived from the image itself, which is included in the processing chain repository.

Methodology needs to be restructured. First, please provide a flowchart to give an overview of all the major processing steps outlined in this section. This can help readers understand the general processing workflow. Second, there are many nice sketch figures presented in the method section, while they are helpful for readers to visualize each processing step, most of them can be easily merged into one single integrated figure, such as Figures 2/3/5/7. For Figure 5 and Figures 7B-E, I not even sure whether they are necessary in the main text as these details are trivial concepts that are easy to understand from the text description alone, perhaps it's better to put them in the supplementary material.

- Thank you for this helpful feedback. We have included a flowchart of the processing chain (Figure 2)
- We combined Figures 2 and 3 into a single figure (now Figure 3), moved Figure 7 to the supplementary (now Figure S2), and removed Figure 5 from the manuscript.

Khan (2017) dataset: in line 128 it says the time period of Khan dataset is 1995-2015. However, from the data link you provided in the manuscript (https://dataverse.geus.dk/dataset.xhtml?persistentId=doi:10.22008/FK2/GQJJEA), the Khan data covers the period from 2011 to 2020, which is correct? The more urgent issue is why not use the latest Khan (2023) dhdt dataset? Won't this latest dhdt improve the ice thickness calculation in this study? Again, there is no introduction of Khan dhdt dataset in the Data Sources section. Why did you choose to use Khan dhdt instead of other dhdt products? What is the advantage of this data product and how it was generated? Please also cite the associated publication for the Khan dataset.

- We now refer only to the Khan (2023) dataset in the manuscript, which contains 2 datasets: one covering 1995-2015, and one covering 2011-2020. We have merged these two datasets into a single dataset covering 1995-2020. This is now clarified in the text.
- We note that part of the confusion here was due to data migration to the GEUS server and the extension of the product: Khan (2017) only had 1995-2015, but the new version adds a dataset covering 2011-2020.

In Figure 6, if Khan dhdt is already available for the date of the terminus change, doesn't this mean that TOD has already been included in the K-SCR time range? Why there is still an additional step of checking whether TOD is in K-SCR time range?

- We amended and simplified the figure to make the processing steps easier to follow (now Figure 4).

**ArcticDEM and AeroDEM:** what type of ArcticDEM is used in the methods? Are they mosaics or strips? If using the strip and it covers the selected terminus trace, is it still necessary to adjust elevation based on dhdt?

- We use ArcticDEM strips, downloaded via GoogleEarthEngine from the University of Minnesota Polar Geospatial Center. We have added further information on this to the manuscript, including the spatial resolution and acquisition type to the description of the DEMs [Lines 118-121] and to Table 1.

- We calculate surface elevation for each individual terminus position i.e. timestep. Unless the acquisition date of the ArcticDEM coincides exactly with the date of the terminus position, we then adjust for surface elevation change using dhdt (when available).

Figure 9 gives an overview of the frontal ablation rates produced in this study. The choices of different number scales of the annual mean frontal ablation look a bit random, why not use integers here or a colorbar that can clearly show the value variations? Can you also compare the frontal ablation rates with the Mankoff ice discharge data statistically, ideally in a histogram? This can show the impact and necessity of including terminus changes in the frontal ablation calculation.

- We simplified the former Figure 9 (now Figure 6) to make the varying frontal ablation estimates between individual glaciers clearer.
- We have included a comparison plot between the frontal ablation estimates shown in this study and Kochtitzky et al (2023) in Figure 6 B, as well as a comparison figure of the ice discharge from Mankoff et al. (2020) and Kochtitzky et al. (2023) in Figure 6 C.
- We have added a chart of how ablation compares to discharge for each glacier in Figure S3.

Figure 10 and Line 380-383. Authors claimed that "agreement is reduced for the period 2010-2020", this is not obvious from Figure 10, although Table S3 provides a comparison. I recommend calculating the difference in frontal ablation for each glacier between this study and Kochtitzky et al. (2023) dataset, then plotting these differences in a map similar to Figure 9, or in a histogram.

- Upon further detailed analysis the two periods appear to have comparable agreement and we have removed this statement from the main text.

Line 398-400, it briefly touched on the issue of ice discharge but only gave one example. Can you compare the discharge data used in Kochtitzky et al. (2023) and the Mankoff ice discharge data for all the studied glaciers? This should help clarify if the discrepancy in frontal ablation rates is from terminus change or ice discharge.

- Thank you for this suggestion. We have included a comparison of ice discharge between the two studies which hopefully clarifies this point (Figure 6 C).
- It remains difficult, though, to determine which discharge product is more accurate without further analysis, which would go beyond the scope of this paper.

Line 346: This sharp increase in frontal ablation (Figure 8A) started in 2005/2006, not 2004/2005. There is significant interannual variability in frontal ablation and given there is no errorbar provided in the plot, it is difficult to claim that this increase in frontal ablation is from changes in velocity and terminus, especially since there is also a sharp change in 2010/2011 according to Figure 8A.

- We appreciate this comment and have substantially revised this paragraph. We now focus on the increase in frontal ablation variability and how it coincides with changes in velocity and terminus position. We use this example to validate our approach, since Helheim glacier has been studied extensively and the 2005/6 retreat is well documented in the literature. We are now careful to avoid discussing causal relations.

Line 348-351: can you fit linear regression before and after 2004/2005 to see if there is actually an increase in frontal ablation before and after the velocity/terminus changes?

- We appreciate this suggestion – there is a small increase in linear slope but it's unlikely statistically significant, due to the high variability in the time series. As per the comment above, we are now no more suggesting an increase in frontal ablation but rather focus on the increase in frontal ablation variability.

In Figure 8 and Figures S2-S48, the temporal intervals in the x-axis are not regular – it contains both two-year and three-year intervals but they have the same width, is this a plotting error?

- We have amended this oversight and the intervals between dates are now consistently 5 years.

**Technical comments**

Abstract needs rewrite. For example, BedMachine has been mentioned twice and please be clear in Line 34 this is consecutive *calving front* observations.

- We have rewritten the abstract to avoid duplicate mentions of datasets and to make it clearer that this manuscript describes a data product rather than a software product.

Line 38-39: "any tidewater glacier". Is it only for the Greenland Ice Sheet? Or globally?

- We clarified that the processing chain can be adapted for all tidewater glaciers globally [Lines 38-40 and Lines 473-475].

1. Line 69-73: this paragraph reads like a separate statement and feels very sudden here. Since this is a dataset paper for a dataset journal, the focus be the data product itself instead of the methodology, please recheck the journal submission guideline.

   - We removed the paragraph and added a more extensive description of the data product [Lines 83-106].

2. Line 105-106: could you please provide a figure on the spatial coverage of the Bedmachine v4 bathymetry data sources in the supplementary file?

   - We appreciate this suggestion. The BedMachine v4 dataset is widely used in in the field, openly accessible, and linked in the data sources section. While we agree that this article should be largely self-contained, we'd argue that the ubiquity of the BedMachine v4 product alleviates the need for a figure.

3. Line 127: as mentioned in the major comments, please provide detailed information about the Khan dhdt dataset here.

   - Please see our response to the major comment above. We added further details on the Khan (2023) surface elevation change datasets to the paragraph [Lines 122-125 and Lines 312-317] and Table 1.

4. Line 142: "5 km upstream of the terminus" is this the most retreated terminus location?

   - The distance of the flux gate from the terminus was chosen by Mankoff et al. (2020). We manually measured the distance of each flux gate to the most retreated position to determine whether we need to adjust surface elevation as there might be a time lag between the gate and the terminus.

- The flux gates are, on average, 5 kilometres upstream from the most retreated terminus position.

5. Line 173: please be clear that the individual terminus positions are compared to each other to get area changes.

   - Thank you. We rephrased the sentence to [Lines 287-288]: "*A reference boundary needs to be defined so that area change can be calculated by comparing individual terminus positions to each other".*

6. Line 182 **Terminus Positions**: I suggest introducing terminus positions first before talking about fjord boundary, because in fjord boundary section there are lots of descriptions on how to change the terminus direction without knowing what terminus data were used.

   - We agree that the fjord geometry section does refer to the terminus position data at times. However, we would like to keep the section on fjord boundaries at its current location in the manuscript, as this is relevant for the subsequent section on processing terminus delineations.
   - We moved the section on defining the upstream boundary below the section on terminus positions (now section 2c)

7. Line 188-191: this sentence is difficult to understand, please rephrase. Can you give a number on the maximum level of uncertainty involved?

   - We rephrased the sentence to clarify the meaning. It now reads [Lines 205-209]: "*We found that using terminus positions spaced at closer than 1 month gives unreliable frontal ablation estimates (the error increases as the time interval decreases – see discussion of errors below). We also found that delineations that are only several days apart and created by different authors can differ significantly and thereby introduce large uncertainties (cf. Goliber et al., 2022)."*
   - Errors for delineations between different authors can be found in Goliber et al. (2022), Figure 12, so that we do not include these uncertainties here but rather refer to the original TermPicks study.

8. Line 191: what is the threshold for time difference?

   - We take the observation that is closest to the 1st of each month. We have aimed to clarify this in the text (Methods section 2b).

9. Line 212: where is this centerline from?

   - The centerlines are manually drawn but are available in the repository for reproducibility of the study. We've included a statement in the sentence, which now reads [Lines 238-241]: "*Ice flow velocities are successively extracted along a centerline, which has been drawn manually for each glacier (available in the repository), between the most retreated and most advanced terminus position for each glacier.* "

10. Line 217: change "mean value velocity" to mean velocity

    - Changed as suggested.

11. Line 229-232: please move this information to Line 103-110 when first mentioning how these 49 glaciers were selected.

- We have added a sentence to the product description to clarify that some glaciers were excluded [Lines 1041068].

12. Line 269: please be clear about "the SCR", is this "AA-SCR"?

- Changed to AA-SCR as suggested.

13. Line 357-359: please be more explicit about the importance of having this higher variability in frontal ablation. Or why does it matter to have this variability compared to ice discharge that doesn't have it?

- We included two sentences to discuss why having a higher variability is important [Lines 433-440].

14. Figure 2: please label the upper and lower fjord walls in the figure.

- We labelled the fjord boundaries in what is now Figure 3 and have amended the text to use fjord boundaries 1 and 2 (rather than upper/lower).

15. Figure 3 and Line 204-205: Can you explain a bit about how these delineations can skew the mass changes? It is not clear from Figure 3 at all.

- We added clarification on how shorter delineations could skew the mass change calculations to the sentence. The sentence now reads [Lines 231-233]:
  *"[…] as extrapolation of these delineations to the fjord boundaries would create an arbitrary terminus geometry which could subsequently skew mass change calculations (Fig. 3D)."*

16. Figure 8: In Line 351-355, the figure labels are all wrong here, and there is no Figure 8E, please carefully check the figure labels throughout the manuscript. It is hard to distinguish between TMC and F in Figure 8A given the current choices of linewidth and line color. In Figure 8C, which direction is terminus retreat or advance?

- We thank the reviewer for spotting these issues and have changed the labels in the text to correspond with the panels in the figure.
We changed the colors and linewidths in the Figure 8 (now Figure 5) and all supplementary figures (S4-S54) to make them more distinguishable